# DynVLA: Learning World Dynamics
# for Action Reasoning in Autonomous Driving

**Shuyao Shang** [* 1 2]   **Bing Zhan** [* 1 2]   **Yunfei Yan** [1 2]   **Yuqi Wang** [1 2]   **Yingyan Li** [1 2]   **Yasong An** [3]
**Xiaoman Wang** [3]   **Jierui Liu** [3]   **Lu Hou** [3]   **Lue Fan** [1 2 ✉]   **Zhaoxiang Zhang** [1 2 ✉]   **Tieniu Tan** [1 4]

## Abstract

We propose DynVLA, a driving VLA model that introduces a new CoT paradigm termed Dynamics CoT. DynVLA forecasts compact world dynamics before action generation, enabling more informed and physically grounded decision-making. To obtain compact dynamics representations, DynVLA introduces a Dynamics Tokenizer that compresses future evolution into a small set of dynamics tokens. Considering the rich environment dynamics in interaction-intensive driving scenarios, DynVLA decouples ego-centric and environment-centric dynamics, yielding more accurate world dynamics modeling. We then train DynVLA to generate dynamics tokens before actions through SFT and RFT, improving decision quality while maintaining latency-efficient inference. Compared to Textual CoT, which lacks fine-grained spatiotemporal understanding, and Visual CoT, which introduces substantial redundancy due to dense image prediction, Dynamics CoT captures the evolution of the world in a compact, interpretable, and efficient form. Extensive experiments on NAVSIM, Bench2Drive, and a large-scale in-house dataset demonstrate that DynVLA consistently outperforms Textual CoT and Visual CoT methods, validating the effectiveness and practical value of Dynamics CoT. Project Page: https://yaoyao-jpg.github.io/dynvla/

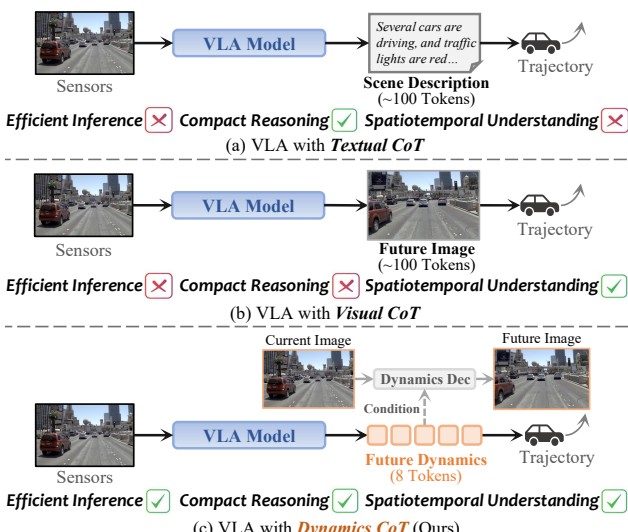

*Figure 1.* **Comparison of different CoT paradigms in autonomous driving VLA models.** (a) Textual CoT suffers from limited spatiotemporal understanding and high inference latency due to long textual reasoning traces. (b) Visual CoT introduces substantial redundancy and computational overhead from pixel-level generation. (c) Dynamics CoT compresses future dynamics into a small set of tokens, achieving latency-efficient inference with compact reasoning and accurate spatiotemporal modeling.

## 1. Introduction

End-to-end autonomous driving (Hu et al., 2023; Sima et al., 2025; Song et al., 2025; Zhang et al., 2025a; Xing et al., 2025b; Hu et al., 2025) has recently embraced the Vision–Language–Action (VLA) paradigm (Huang et al., 2025; Yang et al., 2025a), which offers richer cross-modal grounding and improved decision quality. Inspired by the cognitive process where human drivers reason causally before acting, VLA models also benefit from this "reason-then-act" process, making Chain-of-Thought (CoT) (Wei et al., 2022) well-suited for driving tasks. By reasoning about why certain maneuvers are required in interaction-intensive and rule-constrained traffic, VLA models can produce more reliable driving decisions in long-horizon, safety-critical scenarios.

Within VLA-based autonomous driving, the dominant de-

[1]New Laboratory of Pattern Recognition (NLPR), Institute of Automation, Chinese Academy of Sciences [2]School of Artificial Intelligence, University of Chinese Academy of Sciences [3]Yinwang Intelligent Technology Co. Ltd. [4]Nanjing University. Correspondence to: Lue Fan <lue.fan@ia.ac.cn>, Zhaoxiang Zhang <zhaoxiang.zhang@ia.ac.cn>.

sign for CoT is Textual CoT (Fig. 1a), which performs reasoning in the text space and provides high-level decision logic (Hwang et al., 2024; Li et al., 2025b). However, driving maneuvers depend on fine-grained spatiotemporal relationships in a complex, physically constrained world, which discrete linguistic representations struggle to capture. To address this issue, recent works have explored Visual CoT (Fig. 1b), which predicts future visual frames and subsequently generates actions, enabling spatiotemporal reasoning in the pixel space (Zeng et al., 2025; Zhao et al., 2025b). Although Visual CoT is more expressive in representing spatiotemporal relationships, the model must predict both decision-irrelevant background and texture-level details, which increases reasoning redundancy and learning difficulty. Furthermore, both Textual and Visual CoT require generating a large number of reasoning tokens, leading to substantial inference latency.

To overcome these limitations, we propose **DynVLA**, which introduces a new CoT paradigm termed **Dynamics Chain-of-Thought** (Dynamics CoT). DynVLA first compresses future dynamics into compact tokens, and then predicts these dynamics tokens before action generation. Compared to Textual CoT, Dynamics CoT models the evolution of spatiotemporal states beyond symbolic textual reasoning. Compared to Visual CoT, it avoids redundant reasoning by encoding scene dynamics only. In addition, this compact representation models the state transition between consecutive observations, and therefore only requires a small number of tokens to capture future dynamics. This substantially shortens the reasoning trace and reduces inference latency by over an order of magnitude compared to Textual or Visual CoT.

To represent such dynamics in a compact and learnable form, prior studies have explored latent action tokenizer for embodied scenarios (Ye et al., 2024). However, driving scenes involve more pronounced ego-viewpoint transformations and richer dynamics from multiple interacting agents. To address this gap, we introduce a Dynamics Tokenizer tailored for driving scenarios. It first factorizes dynamics into two decoupled factors: ego-centric dynamics, arising from the ego vehicle's own motion, and environment-centric dynamics, originating from external changes such as other traffic participants. However, dynamics cannot be naturally disentangled, and the learned representation may become physically ambiguous. For instance, ego forward motion can be confused with a leading vehicle moving backward. We therefore introduce physical regularization to align ego-centric dynamics with ego motion. In addition, we note that different views (e.g., image and BEV) should share the same underlying dynamics representation. We thus introduce cross-view consistency regularization, leading to semantically aligned dynamics for planning. Finally, following the standard training strategy of CoT-based models,

we apply supervised fine-tuning (SFT) and reinforcement fine-tuning (RFT) on Dynamics CoT, enabling reasoning in the dynamics space and improving decision quality.

We conduct comprehensive evaluations on a real-world benchmark NAVSIM (Dauner et al., 2024), a closed-loop benchmark Bench2Drive (Jia et al., 2024), and a massive in-house dataset. Experimental results demonstrate that Dynamics CoT outperforms both non-CoT VLA methods and Textual CoT or Visual CoT methods, validating its effectiveness and practical value in autonomous driving.

The contributions of this paper can be summarized as follows: (1) We propose DynVLA, which introduces a new CoT paradigm called Dynamics CoT for autonomous driving VLA models. Dynamics CoT reasons over compact future dynamics that both capture spatiotemporal evolution and reduce reasoning redundancy. (2) We identify that naive dynamics tokenization tends to entangle ego dynamics and environment dynamics, and address this by explicitly decoupling them with physically grounded regularization. (3) We visualize the transferability of learned dynamics tokens and conduct extensive experiments and empirical analyses across multiple benchmarks, demonstrating the effectiveness of Dynamics CoT.

## 2. Related Works

### 2.1. VLA models for End-to-End Autonomous Driving

Vision-Language-Action (VLA) models have proven effective in robotics (Kim et al., 2024; Zitkovich et al., 2023; Black et al., 2024; 2025; Wang et al., 2025b; Zheng et al., 2025), and recent work has increasingly transferred this paradigm to end-to-end autonomous driving. Early explorations primarily adopt an LLM backbone (Mao et al., 2023; Shao et al., 2024a; Xu et al., 2024), while more recent systems move toward VLM-based policies (Zhou et al., 2024; Huang et al., 2024; Zhou et al., 2025a). Beyond directly regressing trajectories, DiffVLA (Jiang et al., 2025) proposes VLM-guided diffusion planning for multi-modal trajectory generation, and ORION (Fu et al., 2025a) addresses the mismatch between semantic reasoning space and continuous action space with a generative planner. To better inject domain knowledge from driving, ReCogDrive (Li et al., 2025d) introduces a hierarchical pipeline that distills human driving cognition into the VLM, while DriveVLA-W0 (Li et al., 2025a) leverages world-model pretraining to provide dense supervision.

### 2.2. Chain-of-Thought in VLA models

Chain-of-Thought (CoT) (Wei et al., 2022) is known to improve the performance of LLM by introducing extra thinking steps before producing final answers (Chen et al., 2025a; Wang et al., 2025c; Xia et al., 2025; Sun et al., 2024), and

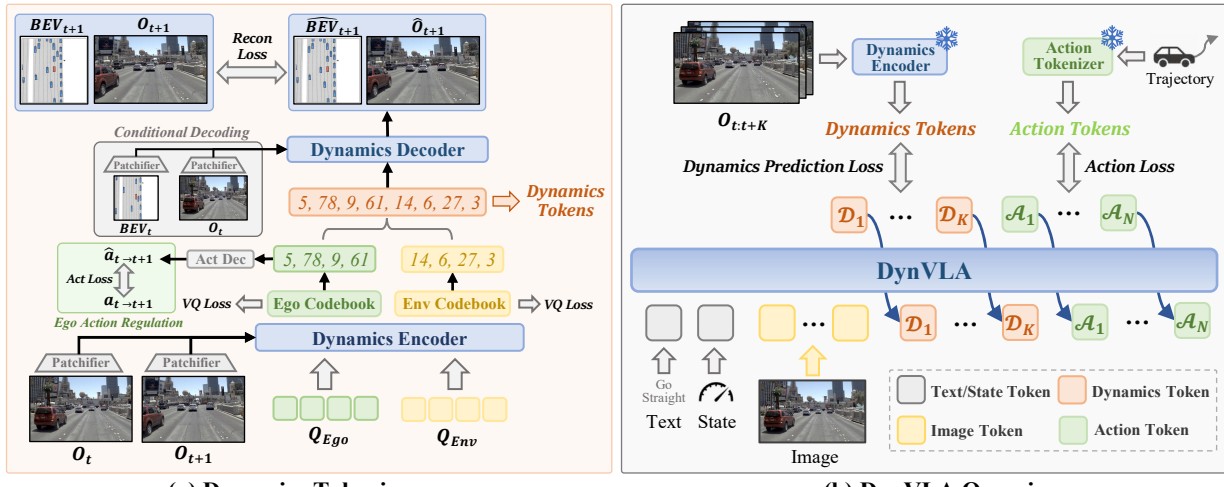

**(a) Dynamics Tokenizer**  **(b) DynVLA Overview**

*Figure 2.* **Overview of the DynVLA.** (a) Given adjacent image observations, a dynamics encoder extracts ego-centric and environment-centric dynamics, which are discretized via VQ codebooks. Then, the ego-centric dynamics are regularized by the GT ego action, and the combined dynamics are decoded to reconstruct the future image and BEV map conditioned on each current state. (b) DynVLA is supervised to first generate discrete dynamics tokens followed by action tokens, forming structured Dynamics CoT modeling.

robotics research has explored introducing CoT into VLA models. ECoT (Zawalski et al., 2024) explicitly produces structured textual reasoning grounded in the physical environment, while OneTwoVLA (Lin et al., 2025) further unifies fast control and slow reasoning by adaptively invoking CoT. Recent works introduce Visual CoT (Zhao et al., 2025a; Lv et al., 2025), which synthesizes goal-conditioned future images and subsequently generates actions. VLA in autonomous driving also incorporates CoT-style reasoning (Wang et al., 2025a; Xing et al., 2025a). EMMA (Hwang et al., 2024) recasts multiple driving tasks into language space and can generate trajectories together with language-based reasoning. AutoDrive-$R^2$ (Yuan et al., 2025) incentivizes reasoning and self-reflection by combining CoT with reinforcement learning. AutoVLA (Zhou et al., 2025c) proposes adaptive reasoning to reduce unnecessary CoT while preserving it in complex scenarios. Following the recent success of world models in autonomous driving (Zhou et al., 2025b; Liang et al., 2025), FSDrive (Zeng et al., 2025) introduces a world-model-based Visual CoT and generates future visual states as intermediate reasoning steps. In contrast, we focus on a compact dynamics representation that captures spatiotemporal relationships while avoiding redundant generation, enabling efficient planning.

## 3. Method

In this section, we present DynVLA, as illustrated in Fig. 3. DynVLA first trains a Dynamics Tokenizer to extract discrete dynamics tokens (Sec. 3.1). It then performs supervised fine-tuning (SFT) on Dynamics CoT sequences (Sec. 3.2) and further refines the VLA policy via reinforce-

ment fine-tuning (RFT) (Sec. 3.3).

### 3.1. Dynamics Tokenizer

**Encoder with Decoupled Dynamics.** Driving scenarios exhibit significant environment dynamics in addition to ego motion. Therefore, we explicitly decouple the dynamics representation into ego-centric and environment-centric tokens. As shown in Fig. 2a, we first map the input images $O_t$ and $O_{t+1}$ into patch sequences $\mathbf{x}_t$ and $\mathbf{x}_{t+1}$ using a ViT patchifier (Dosovitskiy, 2020), and encode $(\mathbf{x}_t, \mathbf{x}_{t+1})$ with a Dynamics Encoder $E_{\mathrm{dyn}}$, which consists of $L_{\mathrm{Enc}}$ Transformer layers (Vaswani et al., 2017). We then introduce two sets of learnable queries, denoted as $Q_{\mathrm{ego}} \in \mathbb{R}^{N_{\mathrm{ego}} \times d}$ and $Q_{\mathrm{env}} \in \mathbb{R}^{N_{\mathrm{env}} \times d}$, where $N_{\mathrm{ego}}$ and $N_{\mathrm{env}}$ denote the number of ego-centric and environment-centric dynamics tokens, and $d$ is the feature dimension. In practice, both $N_{\mathrm{ego}}$ and $N_{\mathrm{env}}$ are kept small. These queries aggregate the dynamics representations as

$$(e_t^{\mathrm{ego}}, e_t^{\mathrm{env}}) = E_{\mathrm{dyn}}(\mathbf{x}_t, \mathbf{x}_{t+1}; Q_{\mathrm{ego}}, Q_{\mathrm{env}}), \quad (1)$$

where $e_t^{\mathrm{ego}} \in \mathbb{R}^{N_{\mathrm{ego}} \times d_{\mathrm{VQ}}}$ and $e_t^{\mathrm{env}} \in \mathbb{R}^{N_{\mathrm{env}} \times d_{\mathrm{VQ}}}$ denote the continuous ego-centric and environment-centric dynamics representations, and $d_{\mathrm{VQ}}$ is the codebook feature dimension. We maintain two separate VQ codebooks for the ego and environment branches denoted as $\mathcal{C}_{\mathrm{ego}} = \{c_i^{\mathrm{ego}}\}_{i=1}^{M_{\mathrm{ego}}}$ and $\mathcal{C}_{\mathrm{env}} = \{c_j^{\mathrm{env}}\}_{j=1}^{M_{\mathrm{env}}}$, where each code $c_i^{\mathrm{ego}}, c_j^{\mathrm{env}} \in \mathbb{R}^{d_{\mathrm{VQ}}}$, and $M_{\mathrm{ego}}$ and $M_{\mathrm{env}}$ denote the codebook sizes. Then, continuous dynamics $e_t^{\mathrm{ego}}$ and $e_t^{\mathrm{env}}$ are discretized via nearest-neighbor codebook assignment (Van Den Oord et al., 2017), producing discrete tokens $\mathcal{D}_t^{\mathrm{ego}}$ and $\mathcal{D}_t^{\mathrm{env}}$, which are concatenated

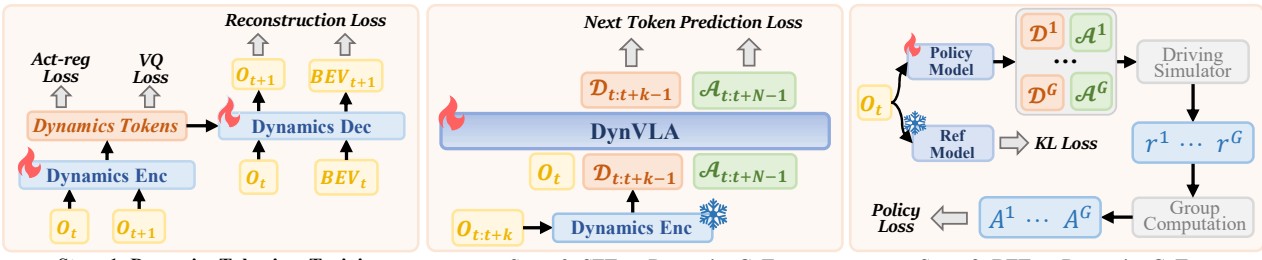

**Stage 1: Dynamics Tokenizer Training**    **Stage 2: SFT on Dynamics CoT**    **Stage 3: RFT on Dynamics CoT**

*Figure 3.* **Overview of the training pipeline for DynVLA**. DynVLA first learns a Dynamics Tokenizer by reconstructing future states from adjacent frames, producing discrete dynamics tokens. It then performs SFT on Dynamics CoT, training the model to generate dynamics tokens before action tokens. Finally, the policy is optimized via RFT with trajectory-level reward and KL regularization.

to finally form the Dynamics Tokens as

$$\mathcal{D}_t = [\mathcal{D}_t^{\text{ego}}, \mathcal{D}_t^{\text{env}}]. \tag{2}$$

The discrete tokens in $\mathcal{D}_t$ are subsequently mapped back to their continuous embeddings via codebook lookup, yielding $z_t \in \mathbb{R}^{(N_{\text{ego}}+N_{\text{env}}) \times d_{\text{VQ}}}$, which serves as the input to the dynamics decoder.

**Decoder with Action-based Regularization.** After obtaining encoded dynamics features, we employ a decoder to reconstruct future observations, providing supervision for learning the Dynamics Tokens. During decoding, we also condition on the current observation, which relieves the discrete tokens from encoding static background or texture details. However, learning dynamics purely from reconstruction remains under-constrained and can lead to codebook collapse. To address this, we introduce an action-based regularization that aligns ego-centric dynamics with the ego motion. Specifically, we use a two-layer MLP action decoder to predict the ego action $\hat{\mathbf{a}}_{t \to t+1}$ from the ego-centric dynamics $\mathcal{D}_t^{\text{ego}}$ and penalize the discrepancy between the predicted action and the ground-truth action by $\mathcal{L}_{\text{act-reg}} = \|\hat{\mathbf{a}}_{t \to t+1} - \mathbf{a}_{t \to t+1}\|_2^2$. This encourages the ego-centric branch to explicitly explain ego motion, thereby promoting disentangled learning between ego-centric and environment-centric dynamics.

**Decoder with Cross-view Consistency Regularization.** We note that desirable environment dynamics should capture the same underlying scene evolution across different representations. Therefore, we impose a cross-view consistency regularization by requiring the same Dynamics Tokens to predict both the future image and the future BEV map conditioned on their respective current observations. This enforces semantic consistency between the image and BEV spaces, yielding more coherent environment-centric dynamics. As shown in Fig. 2a, we map the current image $O_t$ and the current BEV map $BEV_t$ into patch sequences $\mathbf{x}_t$ and $\mathbf{b}_t$ via a ViT patchifier, and then perform conditional decoding with two modality-specific dynamics decoders: an image

decoder $D_{\text{dyn}}^{\text{img}}$ and a BEV decoder $D_{\text{dyn}}^{\text{bev}}$, each composed of $L_{\text{Dec}}$ Transformer layers. Both decoders are conditioned on the same dynamics representation $z_t$ and their respective patch sequences, and each predicts its corresponding future state as

$$\widehat{O}_{t+1} = D_{\text{dyn}}^{\text{img}}(\mathbf{x}_t, z_t), \quad \widehat{BEV}_{t+1} = D_{\text{dyn}}^{\text{bev}}(\mathbf{b}_t, z_t). \tag{3}$$

**Dynamics Tokenizer Training.** The Dynamics Tokenizer is trained by minimizing the reconstruction loss, the VQ-VAE loss, and the regularizations loss. The image reconstruction loss $\mathcal{L}_{\text{recon}}^{\text{img}}$ combines a mean squared error loss and a perceptual similarity loss (Zhang et al., 2018) to capture both low-frequency structural consistency and high-level semantic similarity, while the BEV reconstruction loss $\mathcal{L}_{\text{recon}}^{\text{bev}}$ is a cross-entropy loss. The overall training objective of the Dynamics Tokenizer is given by

$$\mathcal{L} = \mathcal{L}_{\text{recon}}^{\text{img}} + \lambda_{\text{bev}}\mathcal{L}_{\text{recon}}^{\text{bev}} + \lambda_{\text{vq}}\mathcal{L}_{\text{VQ}} + \lambda_{\text{act-reg}}\mathcal{L}_{\text{act-reg}}, \tag{4}$$

where $\mathcal{L}_{\text{VQ}}$ is the vector-quantization loss (Van Den Oord et al., 2017), and $\lambda_{\text{bev}}, \lambda_{\text{vq}}, \lambda_{\text{act-reg}}$ are the corresponding weighting coefficients.

### 3.2. SFT on Dynamics CoT

**Structured Dynamics CoT Sequence.** To realize Dynamics CoT, we perform supervised fine-tuning (SFT) on structured dynamics sequences (as shown in Fig. 2b). Given the current and next $K$ frame images $O_{t:t+K}$ and letting $E_{\text{dyn}}$ denote the trained dynamics encoder, the dynamics tokens at step $t + k$ are defined as

$$\mathcal{D}_{t+k} = E_{\text{dyn}}(O_{t+k}, O_{t+k+1}), \quad 0 \le k \le K - 1, \tag{5}$$

where each $\mathcal{D}_{t+k}$ consists of $N_{\text{ego}} + N_{\text{env}}$ discrete dynamics tokens. To explicitly delineate the dynamics reasoning sequence, we introduce two special tokens $\langle\text{BOD}\rangle$ and $\langle\text{EOD}\rangle$, which mark the beginning and end of dynamics reasoning. For action generation, we encode the continuous action into a discrete action token sequence $\mathcal{A}_{t:t+N-1}$ using the FAST tokenizer (Pertsch et al., 2025), where $N$ denotes the length

of the action token sequence. Similarly, we introduce $\langle\text{BOA}\rangle$ and $\langle\text{EOA}\rangle$ to indicate the beginning and end of the action generation sequence. For each training sample, the target output sequence is organized as

$$\mathbf{y} = [\langle\text{BOD}\rangle, \mathcal{D}_{t:t+K-1}, \langle\text{EOD}\rangle, \langle\text{BOA}\rangle, \mathcal{A}_{t:t+N-1}, \langle\text{EOA}\rangle]. \quad (6)$$

**SFT Training.** Given the target sequence, we train the model by maximizing the likelihood of the output tokens conditioned on the observation and instruction context. Specifically, the model input at time $t$ is denoted as $\mathbf{c}_t = \{O_t, O_{t-1}, T_t, S_t\}$, where $O_t$ is the current image observation, $O_{t-1}$ is the previous image observation, $T_t$ is the text instruction, and $S_t$ represents the ego state. We adopt the standard next-token prediction loss (Vaswani et al., 2017) and minimize the negative log-likelihood over the dynamics reasoning sequence and the action generation sequence, which is formulated as

$$\mathcal{L}_{\text{dyn}} = -\sum_{k=0}^{K-1} \log p_\theta(\mathcal{D}_{t+k} \mid \mathcal{D}_{t:t+k-1}, \mathbf{c}_t), \quad (7)$$

$$\mathcal{L}_{\text{act}} = -\sum_{n=0}^{N-1} \log p_\theta(\mathcal{A}_{t+n} \mid \mathcal{A}_{t:t+n-1}, \mathcal{D}_{t:t+K-1}, \mathbf{c}_t). \quad (8)$$

The overall objective for Dynamics CoT SFT is given by

$$\mathcal{L}_{\text{SFT}} = \mathcal{L}_{\text{dyn}} + \lambda_{\text{act}}\mathcal{L}_{\text{act}}, \quad (9)$$

where $\lambda_{\text{act}}$ is a weighting coefficient. Through this procedure, the pretrained model explicitly learns a causal generation order of dynamics reasoning followed by action generation, and treats the reasoned dynamics as an intermediate variable for decision making.

### 3.3. RFT on Dynamics CoT

While Dynamics CoT SFT teaches the model to explicitly reason future dynamics before acting, the learning of actions remains purely imitation-based. However, imitation learning is prone to generate human-like but unsafe trajectories (Shang et al., 2025) and tends to produce averaged and suboptimal motion plans (Li et al., 2025d). In addition, recent studies have shown that applying reinforcement learning to CoT-based models can provide outcome-driven incentives beyond SFT (Guo et al., 2025). Thus, to address the limitations of imitation learning and in accordance with common practice in reasoning models, we introduce reinforcement fine-tuning (RFT) (Shao et al., 2024b; Guo et al., 2025) to further enhance safety and decision quality.

**Reward Design.** For each trajectory, we adopt the PDM Score (PDMS) (Dauner et al., 2024) as the trajectory-level

reward $r_{\text{traj}}$, which is a scalar in $[0, 1]$. In addition, to stabilize RL training and enforce the model output to follow the CoT template, we introduce a format reward $r_{\text{fmt}} \in \{0, 1\}$, which is 1 if the generated sequence satisfies the required token organization, and 0 otherwise. The final reward is computed as a weighted combination of trajectory reward and format reward: $r = r_{\text{traj}} + \lambda_{\text{fmt}} r_{\text{fmt}}$, where $\lambda_{\text{fmt}}$ is a weighting coefficient.

**RFT Training.** We optimize the policy using Group Relative Policy Optimization (GRPO) (Shao et al., 2024b). For each training sample given $\mathbf{c}_t$, we roll out $G$ candidate sequences $\{o_i\}_{i=1}^G$ and compute their corresponding rewards $\{r_i\}_{i=1}^G$. Then, the GRPO objective can be written as

$$\begin{aligned} \mathcal{J}_{\text{GRPO}}(\theta) = \frac{1}{G}\sum_{i=1}^{G}\frac{1}{|o_i|}\sum_{t=1}^{|o_i|} &\min\Big(\rho_{i,t}(\theta)\,\hat{A}_{i,t}, \\ &\text{clip}\big(\rho_{i,t}(\theta),\,1-\epsilon,\,1+\epsilon\big)\,\hat{A}_{i,t}\Big) \\ &-\beta\,D_{\text{KL}}(\pi_\theta \,\|\, \pi_{\text{ref}}), \end{aligned} \quad (10)$$

where $\hat{A}_i = \frac{r_i - \text{mean}(\{r_j\}_{j=1}^G)}{\text{std}(\{r_j\}_{j=1}^G)}$, $\rho_{i,t}(\theta) = \frac{\pi_\theta(o_{i,t}|\mathbf{c}_t, o_{i,<t})}{\pi_{\theta_{\text{old}}}(o_{i,t}|\mathbf{c}_t, o_{i,<t})}$, $\epsilon$ is the clipping range, $\pi_{\text{ref}}$ is a frozen reference model from SFT, and $\beta$ controls the KL regularization strength. Through GRPO-based RFT, the model can further improve planning safety and decision quality while preserving the structured generation of Dynamics CoT.

## 4. Experiments

### 4.1. Experimental Setup

We conduct comprehensive experiments on three benchmarks: a real-world benchmark NAVSIM (Dauner et al., 2024), a closed-loop benchmark Bench2Drive (Jia et al., 2024), and a large-scale in-house dataset containing 700k frames. Details regarding datasets and evaluation metrics are provided in Appendix A, and implementation details are provided in Appendix B.

### 4.2. Main Results

**NAVSIM Results.** Table 1 reports the performance comparison on the NAVSIM benchmark (Dauner et al., 2024). Among all evaluated methods, DynVLA achieves the highest PDMS, outperforming both traditional end-to-end methods and recent VLA-based methods. Notably, compared with existing VLA methods that use textual or visual CoT, DynVLA yields better planning quality, indicating that reasoning over future dynamics provides more effective and reliable decision-making guidance.

**Bench2Drive Results.** Table 2 presents the results on the Bench2Drive benchmark (Jia et al., 2024), which evaluates

*Table 1.* **Comparison on NAVSIM Benchmark.** The best results are denoted by **bold** and the second best are denoted by underline.

| Method | NC↑ | DAC↑ | TTC↑ | C↑ | EP↑ | PDMS↑ |
|---|---|---|---|---|---|---|
| Human | 100.0 | 100.0 | 100.0 | 99.9 | 87.5 | 94.8 |
| *Traditional End-to-End Methods* | | | | | | |
| VADv2 (Chen et al., 2024) | 97.2 | 89.1 | 91.6 | 100 | 76.0 | 80.9 |
| UniAD(Hu et al., 2023) | 97.8 | 91.9 | 92.9 | 100 | 78.8 | 83.4 |
| TransFuser (Chitta et al., 2022) | 97.7 | 92.8 | 92.8 | 100 | 79.2 | 84.0 |
| PARA-Drive (Weng et al., 2024) | 97.9 | 92.4 | 93.0 | 99.8 | 79.3 | 84.0 |
| LAW (Li et al., 2024b) | 96.4 | 95.4 | 88.7 | 99.9 | 81.7 | 84.6 |
| Epona (Zhang et al., 2025b) | 97.9 | 95.1 | 93.8 | 99.9 | 80.4 | 86.2 |
| Hydra-MDP (Li et al., 2024c) | 98.3 | 96.0 | 94.6 | 100 | 78.7 | 86.5 |
| DiffusionDrive (Liao et al., 2025) | 98.2 | 96.2 | 94.7 | 100 | 82.2 | 88.1 |
| WoTE (Li et al., 2025c) | 98.5 | 96.8 | 94.9 | 99.9 | 81.9 | 88.3 |
| DriveDPO (Shang et al., 2025) | 98.5 | 98.1 | 94.8 | 99.9 | 84.3 | 90.0 |
| *VLA methods w/o CoT* | | | | | | |
| ReCogDrive (Li et al., 2025d) | 98.2 | 97.8 | 95.2 | 99.8 | 83.5 | 89.6 |
| DriveVLA-W0 (Li et al., 2025a) | **98.7** | **99.1** | 95.3 | 99.3 | 83.3 | 90.2 |
| *VLA methods w/ Textual CoT* | | | | | | |
| AutoVLA (Zhou et al., 2025c) | 98.4 | 95.6 | **98.0** | 99.9 | 81.9 | 89.1 |
| AdaThinkDrive (Luo et al., 2025) | 98.4 | 97.8 | 95.2 | 100 | 84.4 | 90.3 |
| AutoDrive-$R^2$ (Yuan et al., 2025) | 98.3 | 94.4 | 95.6 | 100 | 81.6 | 90.3 |
| *VLA methods w/ Visual CoT* | | | | | | |
| FSDrive (Zeng et al., 2025) | 98.2 | 93.8 | 93.3 | 99.9 | 80.1 | 85.1 |
| PWM (Zhao et al., 2025b) | 98.6 | 95.9 | 95.4 | 100 | 81.8 | 88.1 |
| **DynVLA (Ours)** | 98.6 | 98.7 | 95.5 | 100 | **86.8** | **91.7** |

*Table 2.* **Comparison on Bench2Drive Benchmark.** The best results are denoted by **bold** and the second best are denoted by underline. [†] denotes using privileged perceptual information.

| Method | DS↑ | SR↑ | Mean Multi-Ability↑ |
|---|---|---|---|
| Think2Drive[†] (Li et al., 2024a) | 91.85 | 85.41 | 86.26 |
| PDM-Lite[†] (Sima et al., 2024) | 97.02 | 92.27 | 92.82 |
| AD-MLP (Zhai et al., 2023) | 18.05 | 0.00 | 0.87 |
| TCP (Wu et al., 2022) | 40.70 | 15.00 | 14.63 |
| VAD (Jiang et al., 2023) | 42.35 | 15.00 | 18.07 |
| UniAD (Hu et al., 2023) | 45.81 | 16.36 | 15.55 |
| ThinkTwice (Jia et al., 2023b) | 62.44 | 31.23 | 37.17 |
| DriveAdapter (Jia et al., 2023a) | 64.22 | 33.08 | 42.08 |
| Drivetransformer (Jia et al., 2025) | 63.46 | 35.01 | 38.60 |
| Raw2Drive (Yang et al., 2025b) | 71.36 | 50.24 | 53.34 |
| ORION (Fu et al., 2025a) | 77.74 | 54.62 | 54.72 |
| MindDrive (Fu et al., 2025b) | 78.04 | 55.09 | 56.94 |
| AutoVLA (Zhou et al., 2025c) | 78.84 | 57.73 | – |
| TF++ (Jaeger et al., 2023) | 84.21 | 67.27 | 64.39 |
| SimLingo (Renz et al., 2025) | 85.07 | 67.27 | – |
| **DynVLA (Ours)** | **88.34** | **72.73** | **72.23** |

*Table 3.* **Comparison on a large-scale In-house Dataset.** The best results are denoted by **bold**.

| Model | ADE (m)↓ | Collision Rate (‰)↓ |
|---|---|---|
| Transfuser (Chitta et al., 2022) | 1.746 | 5.63 |
| DriveVLA-W0 (VQ) (Li et al., 2025a) | 1.599 | 5.20 |
| DriveVLA-W0 (ViT) (Li et al., 2025a) | 1.344 | 5.13 |
| **DynVLA (Ours)** | **1.215** | **4.04** |

*Table 4.* **Analysis on CoT Design and Latency.** All inference latencies are measured on a single NVIDIA H800 GPU. The best results are denoted by **bold**.

| CoT Content | Latency(s)↓ | NC↑ | DAC↑ | TTC↑ | C↑ | EP↑ | PDMS↑ |
|---|---|---|---|---|---|---|---|
| None (w/o CoT) | **0.20** | 98.3 | 93.8 | 94.6 | 99.9 | 79.5 | 85.6 |
| Scene Description | 3.04 | 98.4 | 93.4 | 94.4 | 99.9 | 79.3 | 85.3 |
| Meta Action | 0.43 | 98.3 | 94.3 | 94.6 | 100 | 79.8 | 86.0 |
| Future Image | 2.29 | **98.7** | 94.4 | 95.0 | 99.9 | 80.0 | 86.3 |
| Optical Flow | 2.29 | 98.6 | 94.4 | 95.3 | 100 | 80.0 | 86.4 |
| **Dynamics (Ours)** | 0.37 | 98.6 | **95.3** | **95.5** | 100 | 80.6 | **87.2** |

*Table 5.* **Ablation on Training Stages.** Dyn CoT denotes Dynamics CoT. The best results are denoted by **bold**.

| Base Model | Dyn CoT | SFT | RFT | NC↑ | DAC↑ | TTC↑ | C↑ | EP↑ | PDMS↑ |
|---|---|---|---|---|---|---|---|---|---|
| EMU3 (Wang et al., 2024) | | ✓ | | 98.3 | 93.8 | 94.6 | 99.9 | 79.5 | 85.6 |
| | ✓ | ✓ | | 98.6 | 95.3 | 95.5 | 100 | 80.6 | 87.2 |
| | | ✓ | ✓ | 98.6 | 96.7 | **95.6** | 100 | 82.3 | 88.7 |
| | ✓ | ✓ | ✓ | 98.6 | 98.7 | 95.5 | 100 | 86.8 | 91.7 |
| Qwen2.5-VL (Bai et al., 2025) | | ✓ | | 98.3 | 93.5 | 94.8 | 100 | 79.1 | 85.3 |
| | ✓ | ✓ | | 98.8 | 94.4 | 95.8 | 100 | 79.9 | 86.6 |
| | | ✓ | ✓ | 98.7 | 96.1 | **96.1** | 100 | 81.6 | 88.4 |
| | ✓ | ✓ | ✓ | 98.8 | 97.9 | 95.9 | 99.9 | 85.8 | 91.0 |

closed-loop driving performance in long-horizon, interactive scenarios. Compared with strong baselines and recent VLA-based methods, DynVLA achieves the best performance across all metrics, demonstrating the advantages of Dynamics CoT in challenging closed-loop environments.

**In-house Dataset Results.** Table 3 reports the results on our large-scale in-house dataset, which is over an order of magnitude larger than public benchmarks, allowing us to evaluate scalability under richer and more diverse driving scenarios. We reimplement a widely adopted end-to-end method Transfuser (Chitta et al., 2022), as well as a strong VLA baseline DriveVLA-W0 (Li et al., 2025a). Compared

to these methods, DynVLA achieves the lowest ADE and Collision Rate, demonstrating more reliable motion prediction and safer maneuver decisions at larger data scales.

### 4.3. Further Analysis and Ablation Studies

**Transferability of Learned Dynamics Tokens.** Fig. 4 presents a demonstration of the transferability of our learned dynamics tokens. Dynamics tokens extracted from one scenario are injected into a new scene and decoded into future states for visualization. The results show that ego-centric dynamics reliably preserve ego motion, while environment-centric dynamics explicitly govern the motions of surrounding agents. Moreover, when both dynamics are combined, the decoded future states accurately reflect the composed dynamics configuration. This demonstrates that the Dynamics Tokenizer learns transferable, disentangled, and interpretable dynamics representations.

**Dynamics CoT Outperforms Other CoT Designs in Both Effectiveness and Efficiency.** We compare different CoT designs and report their inference latency in Table 4. Textual CoT based on scene descriptions incurs substantial inference overhead and degrades performance, suggesting that coarse scene-level descriptions may not provide practical

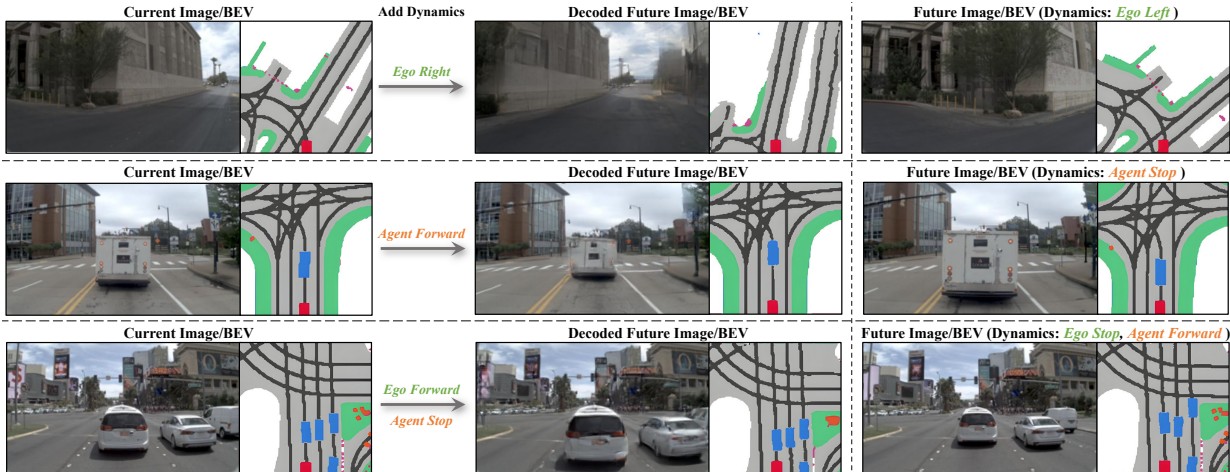

*Figure 4.* **Transferability of learned dynamics.** Dynamics tokens extracted from one scenario are injected into a new scene and decoded into the future image and the BEV map. We contrast the current states, the future states decoded with transferred dynamics, and the original future states. The results show that both ego-centric and environment-centric dynamics are transferable across scenarios.

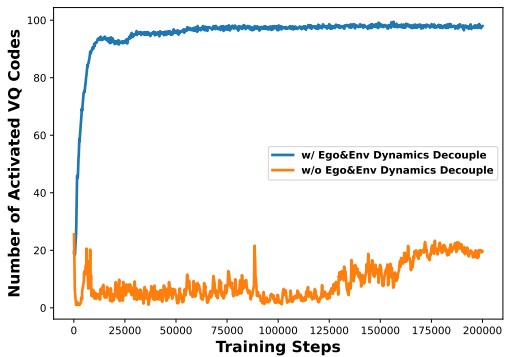

*Figure 5.* **Codebook collapse without dynamics decouple.**

*Table 6.* **Ablation on Dynamics Tokenizer Designs.** Dyn CoT denotes Dynamics CoT. The best results are denoted by **bold**.

| Model | Decouple | Image | BEV | NC↑ | DAC↑ | TTC↑ | C↑ | EP↑ | PDMS↑ |
|---|---|---|---|---|---|---|---|---|---|
| w/o CoT | – | – | – | 98.3 | 93.8 | 94.6 | 99.9 | 79.5 | 85.6 |
| w/ Dyn CoT | | ✓ | ✓ | 98.5 | 94.0 | 94.9 | 100 | 79.5 | 85.8 |
| | ✓ | | ✓ | 98.4 | 94.5 | 94.8 | 100 | 79.8 | 86.2 |
| | ✓ | ✓ | | 98.6 | 94.8 | 95.0 | 100 | **80.6** | 86.7 |
| | ✓ | ✓ | ✓ | **98.6** | **95.3** | **95.5** | 100 | **80.6** | **87.2** |

planning guidance. Reasoning over meta-action decisions yields only marginal performance improvements, suggesting that such high-level symbolic abstractions lack sufficient expressive capacity. Furthermore, visual CoT that predicts future images brings moderate performance gains, but the large number of visual reasoning tokens significantly increases inference latency. We further explore explicit dynamics modeling using current-to-future optical flow as a CoT choice. Although this also improves performance, it remains inferior to ours and still suffers from high latency. Finally, Dynamics CoT achieves the best overall performance without introducing substantial latency overhead.

**Dynamics CoT Strengthens Both SFT and RFT Across Base Models.** Table 5 compares the effects of different training stages under a controlled setting across two base models. Starting from the SFT baseline without CoT, applying Dynamics CoT SFT consistently improves planning-related metrics. In addition, applying RFT to the baseline without CoT also improves PDMS, but the gain is notice-

ably smaller compared with Dynamics CoT RFT. This is because Dynamics CoT provides a compact, structured reasoning trace, enabling RFT to optimize final actions more effectively. Consistent trends across EMU3 (Wang et al., 2024) and Qwen2.5-VL (Bai et al., 2025) also verify the effectiveness and generality of Dynamics CoT. Since EMU3 achieves the best final performance, which we attribute to its unified architecture better fitting the new dynamics modality, we adopt it as the base model for DynVLA.

**Decoupling Dynamics Prevents Codebook Collapse.** Fig. 5 illustrates the number of activated VQ codes during Dynamics Tokenizer training. The number increases rapidly when decoupling dynamics (i.e., introducing separate queries for ego and environment dynamics and applying action-based regularization). However, the tokenizer exhibits an apparent codebook collapse without disentanglement. This is because without decoupling, the tokenizer solely minimizes the reconstruction loss. However, as the decoder conditions on the current observation, much of the background information can be recovered directly, diminishing the necessity for expressive VQ-compressed dynamics representations. By introducing disentangled modeling, the dynamics tokens are forced to learn more discriminative representations, effectively alleviating codebook collapse.

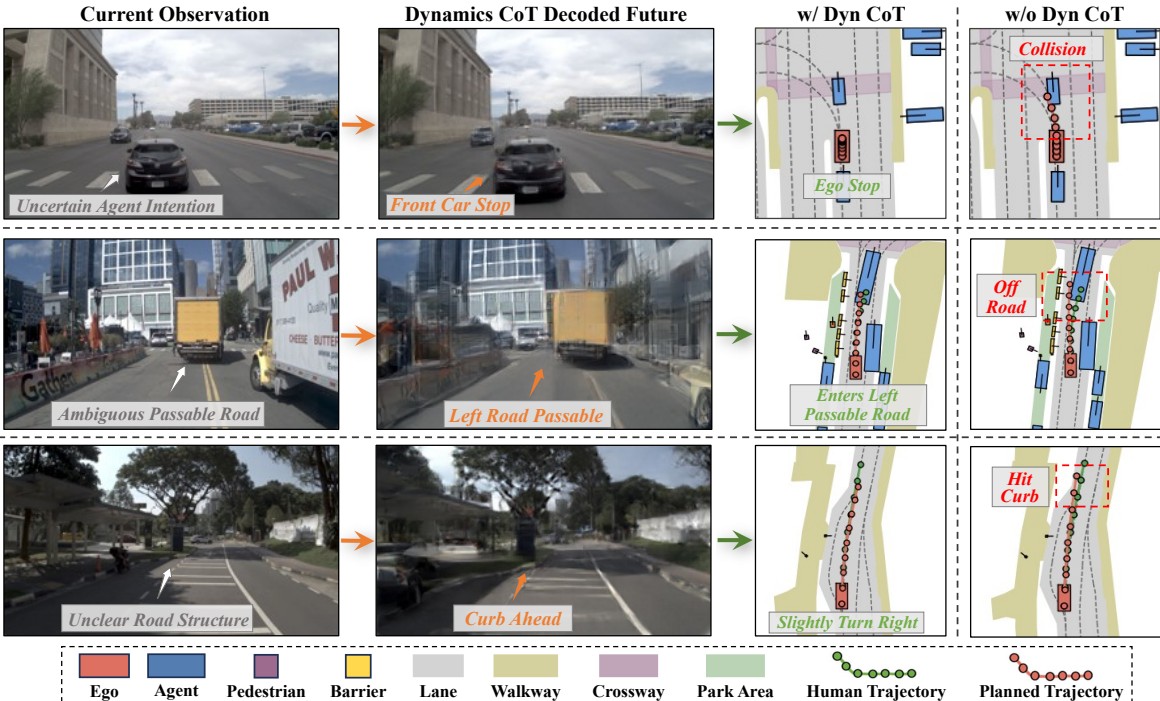

*Figure 6.* **Dynamics CoT improves planning by reasoning over future dynamics.** The first two columns show the current observation and the future decoded by reasoned dynamics. The third and last columns compare planning results with and without Dynamics CoT. Compared to direct action prediction, Dynamics CoT provides intent-aware, foresighted, and constraint-compliant future dynamics, enabling safer and more feasible planning in challenging scenarios.

**Decoupling and Regularization Benefit Planning.** We further analyze the design of the Dynamics Tokenizer in Table 6. The Dynamics Tokenizer without decoupling (i.e., without separate queries or ego-action regularization) yields marginal improvements over the model without CoT, indicating it is unable to capture meaningful dynamics. We also remove either the image branch or the BEV branch, and both result in performance degradation, indicating the importance of cross-view consistency regularization. Finally, combining dynamics decoupling with dual image-BEV supervision achieves the best performance, demonstrating the effectiveness of each component in our design.

### 4.4. Qualitative Analysis

**Safer Intent-aware Interaction.** In interactive driving scenarios, directly committing to action decisions may fail to adjust behavior according to other agents' intentions. However, Dynamics CoT can capture the motion intent of surrounding agents and adjust planning accordingly, leading to safer maneuver decisions. As shown in Fig. 6 (top row), the inferred dynamics indicate that the front car will stop, and the model plans to stop accordingly, avoiding the collision observed without Dynamics CoT.

**Foresighted Planning.** Many planning tasks cannot be directly resolved by instantaneous cues, leading to short-

sighted decisions. In contrast, Dynamics CoT can reason about how traffic evolves over several seconds, revealing future states and enabling foresighted trajectory planning. As shown in Fig. 6 (middle row), Dynamics CoT predicts the leading vehicle moving rightward, which opens a drivable corridor, and the model then exploits this future space to execute a safe maneuver, whereas the model without Dynamics CoT drifts off-road due to a lack of this foresight.

**Road-geometry Awareness.** Beyond agent interactions, driving is also physically constrained by road geometry, yet direct action prediction may struggle to anticipate how such constraints evolve. However, Dynamics CoT can reflect the evolution of road constraints in its predicted dynamics, giving the model awareness of future road boundaries and enabling timely steering adjustments. As shown in Fig. 6 (bottom row), the reasoned dynamics indicate an upcoming curb ahead, guiding the model to slightly turn right and maintain a feasible lane-keeping trajectory, preventing hitting the curb that occurs without Dynamics CoT.

## 5. Conclusion

In this work, we propose DynVLA, which introduces a novel CoT paradigm called Dynamics CoT for VLA-based autonomous driving. Compared to existing CoT methods,

Dynamics CoT reduces redundancy and latency while retaining accurate spatiotemporal understanding. We first introduce a Dynamics Tokenizer that disentangles ego-centric and environment-centric dynamics, and regularize it with ego action supervision and cross-view consistency, enabling physically meaningful and planning-oriented dynamics representations. We then perform SFT on Dynamics CoT to enable explicit reasoning over future dynamics, and further apply RFT to improve decision quality. Experiments across multiple benchmarks demonstrate the effectiveness of DynVLA and highlight Dynamics CoT as a promising direction for reasoning-based VLA models. We further discuss limitations and future works in Appendix E.

## Acknowledgements

This work was supported by New Generation Artificial Intelligence-National Science and Technology Major Project (No. 2025ZD0122902), the Beijing Natural Science Foundation (No. L257004), the National Natural Science Foundation of China (No. 62320106010, No. 62536010, No. 62506356), the Beijing Natural Science Foundation (No. L257015).

## Impact Statement

This paper presents work whose goal is to advance the field of Machine Learning. There are many potential societal consequences of our work, none which we feel must be specifically highlighted here.

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

# Appendix

## A. Datasets and Metrics

**NAVSIM.** NAVSIM (Dauner et al., 2024) is a real-world benchmark providing diverse urban driving data with rich agent interactions. Following standard protocol, we evaluate policy performance using the PDMS, a scalar metric that aggregates multiple safety and efficiency measures. Specifically, PDMS integrates No At-Fault Collision (NC), Drivable Area Compliance (DAC), Ego Progress (EP), Time-to-Collision (TTC), and Comfort (C) as:

$$\text{PDMS} = \text{NC} \times \text{DAC} \times \frac{5 \times \text{EP} + 5 \times \text{TTC} + 2 \times \text{C}}{12}. \quad (11)$$

Here, NC and DAC explicitly gate safety, while EP and TTC capture driving efficiency and temporal risk margins, and C measures comfort.

**Bench2Drive.** Bench2Drive (Jia et al., 2024) is a closed-loop benchmark, which enables scenario-level evaluation under interactive traffic. We report Success Rate (SR), which indicates whether the ego vehicle reaches the intended destination, Driving Score (DS), which additionally accounts for traffic rule penalties, and mean Multi-Ability, which averages the performance across five ability categories: Merging, Overtaking, Emergency Brake, Give Way, and Traffic Sign.

**In-House Dataset.** We further train our model on a large-scale in-house dataset consisting of 700k frames with diverse and balanced distributions over driving scenarios, road structures, actor densities, and agent intentions. Evaluation focuses on safety-critical and long-horizon cases, where we report Average Displacement Error (ADE) and the Collision Rate within 3 seconds. ADE measures the Euclidean displacement between predicted and ground-truth future ego positions, while the Collision Rate quantifies the fraction of predicted trajectories that result in collisions within the horizon.

## B. Implementation Details

**Dynamics Tokenizer** For each driving scene, we use 8 dynamics tokens, consisting of $N_{\text{ego}} = 4$ ego-centric dynamics tokens and $N_{\text{env}} = 4$ environment-centric dynamics tokens, which are inferred from the front-view image. The codebook size for both the ego and environment branches is set to 64, resulting in 128 distinct discrete dynamics token types in total, and the VQ embedding dimension is 32. The Dynamics Tokenizer is implemented with a Transformer architecture. The hidden dimension is set to 1024, the Dynamics Encoder contains $L_{\text{Enc}} = 12$ layers, and both the image decoder and the BEV decoder consist of $L_{\text{Dec}} = 8$ layers. The encoder takes as input the patchified observations $(\mathbf{x}_t, \mathbf{x}_{t+1})$ together with the learned queries $(Q_{\text{ego}}, Q_{\text{env}})$.

We concatenate these four input sequences to form a single token sequence that is fed into the Transformer layers. The ego action used for action regularization corresponds to the relative ego motion between two frames. For BEV supervision, we use the front-view BEV maps provided by the dataset. During training, the image reconstruction loss is composed of an MSE loss and an LPIPS loss, both with a weight of 1.0. The VQ loss weight is set to 1.0, the action regularization loss weight is set to 1.0, and the BEV reconstruction loss weight is set to 0.1. We train the Dynamics Tokenizer for 200k steps on 8 NVIDIA L20 GPUs using a cosine learning rate schedule with 1k warm-up steps and a maximum learning rate of $1 \times 10^{-4}$. The batch size is set to 32, and we use the AdamW optimizer with $\beta_1 = 0.9$ and $\beta_2 = 0.95$.

**Dynamics CoT SFT** We adopt EMU3 (Wang et al., 2024) as the pretrained base model and follow the same pretraining protocol as DriveVLA-W0 (Li et al., 2025a). The Dynamics Tokenizer uses a codebook of 128 discrete dynamics token types, whereas the FAST tokenizer (Pertsch et al., 2025) uses a codebook of 2048 discrete action token types. We replace the last $2048 + 128$ tokens in the token vocabulary with the action tokens and dynamics tokens, respectively. For Dynamics CoT supervision, we extract future dynamics over a 2-second horizon ($K = 2$), yielding a sequence of 16 dynamics tokens that serve as CoT content during SFT. As sensor input, we only use the current front-view image together with the front-view image from 1s earlier. During training, both the Dynamics CoT loss and the action prediction loss are weighted equally with coefficients set to 1.0. We fine-tune the pretrained model for 4k steps on 8 NVIDIA L20 GPUs using a cosine learning rate schedule with 100 warm-up steps and a maximum learning rate of $1 \times 10^{-4}$. The batch size is set to 6, and we use the AdamW optimizer with $\beta_1 = 0.9$ and $\beta_2 = 0.95$.

**Dynamics CoT RFT** We perform reinforcement fine-tuning (RFT) on top of the SFT-trained model. The trajectory reward $r_{\text{traj}}$ and the format reward $r_{\text{fmt}}$ are both weighted with coefficients set to 1.0. RFT is conducted for 6k steps on 6 NVIDIA H800 GPUs using a cosine learning rate schedule with 500 warm-up steps and a maximum learning rate of $2 \times 10^{-6}$. The batch size is set to 6, and the gradient accumulation step is set to 6 as well. The KL coefficient is $1 \times 10^{-3}$ and we use the AdamW optimizer with $\beta_1 = 0.9$ and $\beta_2 = 0.95$.

## C. More Ablation Studies

**Ablation on the Prediction Horizon of Future Dynamics.** Table 7 studies the Dynamics CoT prediction horizon by varying the number of future dynamics steps $K$. Following the formulation in Sec.3, each dynamics token

*Table 7.* **Ablation on the prediction horizon of Dynamics CoT.** The prediction horizon $K$ denotes the number of reasoned future dynamics. The best results are denoted by **bold**.

| Pred Horizon | Latency(s)↓ | NC↑ | DAC↑ | TTC↑ | C↑ | EP↑ | PDMS↑ |
|---|---|---|---|---|---|---|---|
| w/o CoT | **0.20** | 98.3 | 93.8 | 94.6 | 99.9 | 79.5 | 85.6 |
| $K = 1$ | 0.27 | **98.6** | 94.6 | 95.0 | 100 | 80.1 | 86.5 |
| $K = 2$ | 0.37 | **98.6** | **95.3** | **95.5** | 100 | **80.6** | **87.2** |
| $K = 3$ | 0.49 | **98.6** | 94.7 | 95.2 | 100 | 80.3 | 86.7 |
| $K = 4$ | 0.61 | 98.5 | 94.7 | 95.3 | 99.9 | 80.2 | 86.6 |

*Table 8.* **Ablation on the number and allocation of dynamics tokens.** Dyn Num denotes the total number of dynamics tokens, $N_{\text{ego}}$ denotes the number of ego-centric tokens, and $N_{\text{env}}$ denotes the number of environment-centric tokens. The best results are denoted by **bold**.

| Dyn Num | $N_{\text{ego}}$ | $N_{\text{env}}$ | NC↑ | DAC↑ | TTC↑ | C↑ | EP↑ | PDMS↑ |
|---|---|---|---|---|---|---|---|---|
| 8 | 4 | 4 | 98.6 | **95.3** | **95.5** | 100 | **80.6** | **87.2** |
| 8 | 2 | 6 | **98.7** | 94.9 | 95.3 | 100 | 80.5 | 86.9 |
| 8 | 6 | 2 | 98.6 | 94.5 | 95.0 | 100 | 80.2 | 86.4 |
| 4 | 2 | 2 | 98.5 | 94.6 | 95.1 | 100 | 80.1 | 86.4 |
| 16 | 8 | 8 | 98.5 | 94.7 | 94.9 | 100 | 80.3 | 86.5 |

$\mathcal{D}_{t+k}$ encodes the transition between two adjacent images $(O_{t+k}, O_{t+k+1})$, where consecutive steps are spaced by 1s. Compared to the baseline without CoT, introducing Dynamics CoT consistently improves PDMS across all horizons. Increasing the horizon from $K = 1$ to $K = 2$ yields clear gains, indicating that a too short lookahead is insufficient. However, extending the horizon beyond $K = 2$ yields diminishing returns while increasing inference latency. This is because longer horizons introduce greater uncertainty about the future, making the predicted dynamics less reliable and ultimately degrading planning quality. Based on this, we adopt $K = 2$ (corresponding to a 2s future horizon) as the default configuration, which achieves the best performance while maintaining latency-efficient inference.

**Ablation on the Number of Dynamics Tokens.** Table 8 investigates how the capacity and factorization of the Dynamics Tokenizer affect planning performance by varying the total number of dynamics tokens and their allocation to ego-centric versus environment-centric branches. Overall, using 8 dynamics tokens achieves the best PDMS, indicating that a compact yet sufficiently expressive dynamics bottleneck is critical for capturing planning-relevant scene evolution. Reducing the token budget to 4 degrades PDMS, suggesting insufficient capacity to represent simultaneous ego motion and multi-agent interactions. Conversely, increasing the token budget to 16 does not bring further gains and slightly hurts performance, which we attribute to more redundant dynamics representations, ultimately degrading performance. With a fixed budget of 8 tokens, a balanced split ($N_{\text{ego}}$=4, $N_{\text{env}}$=4) outperforms skewed allocations, highlighting that both ego and environment dynamics are neces-

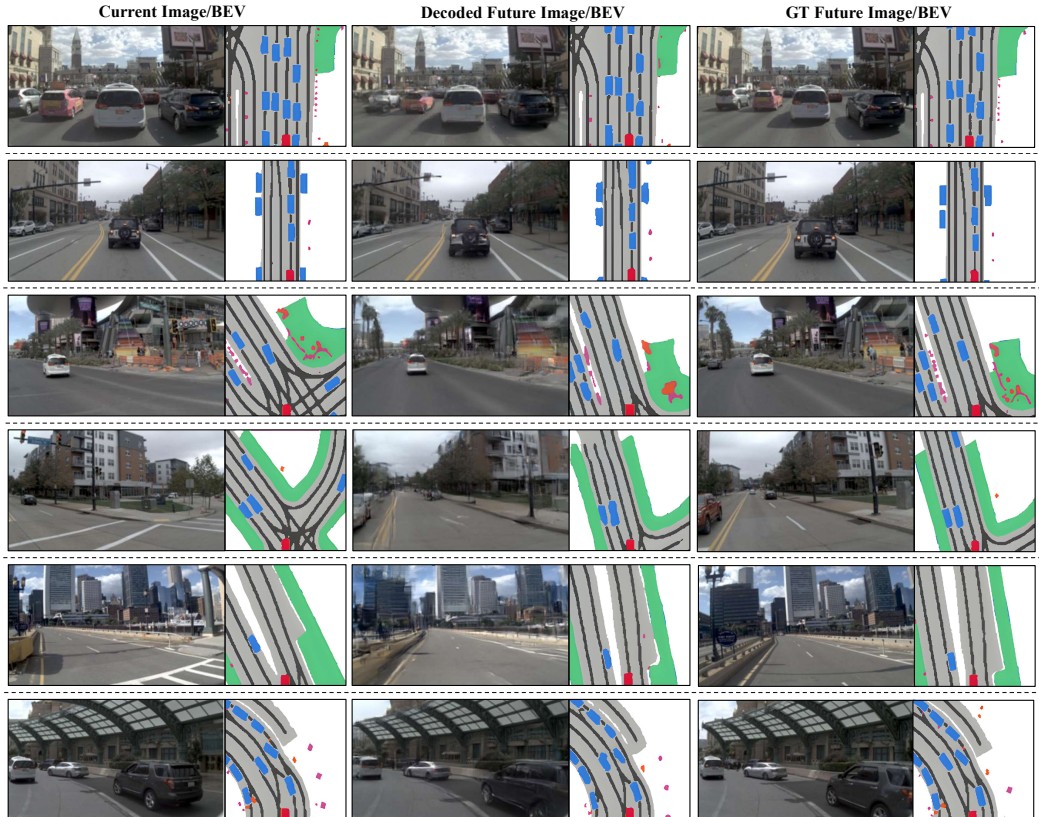

*Figure 7.* **Visualization of the decoded image and BEV from Dynamics Tokenizer.** For each row, we show the current image and BEV map, the decoded future image and BEV map, and the ground-truth future image and BEV map. Across diverse driving scenarios, the decoded futures faithfully capture ego motion and the dynamics of surrounding agents.

sary.

## D. More Qualitative Comparisons

**Decoded Visualization.** We provide a visualization of the decoded outputs from the Dynamics Tokenizer across a diverse set of driving scenarios. As shown in Fig. 7, the decoded futures closely align with the ground-truth evolution in both image space and BEV space, consistently reflecting ego motion and surrounding agent behaviors. These results indicate that the proposed Dynamics Tokenizer effectively compresses planning-relevant ego and agent dynamics.

**Additional Dynamics Transfer Visualizations.** As shown in Fig. 8, we provide additional qualitative visualizations of the cross-scenario dynamics transfer. These results further confirm that the learned dynamics capture disentangled and transferable motion representations that generalize across scenes.

**Additional Qualitative Comparisons with Dynamics CoT.** As shown in Fig. 9, we provide additional qualitative examples demonstrating the effect of Dynamics CoT on planning.

Compared to models without Dynamics CoT, incorporating dynamics reasoning enables the policy to anticipate future scene evolution, thereby improving decision-making.

## E. Limitations and Future Works

Although Dynamics CoT provides structured intermediate reasoning that benefits safety-critical planning, incorrect reasoning traces may induce suboptimal decisions. Similar to reasoning models where flawed CoT leads to erroneous final answers, Dynamics CoT may generate inaccurate future dynamics under highly complex or uncertain driving scenarios, propagating these errors into the subsequent action generation stage. A promising future direction is to enhance Dynamics CoT with richer driving domain priors and world knowledge. In particular, integrating structured world-model knowledge, rule-based priors, or map-aware commonsense may improve the fidelity of predicted dynamics tokens. Another promising direction is to cast our dynamics CoT as the slow component in a fast-slow dual-system driving architecture (Tian et al., 2024; Mei et al., 2024): the slow module performs long-horizon dynamics reasoning and updates dynamics tokens at a lower frequency,

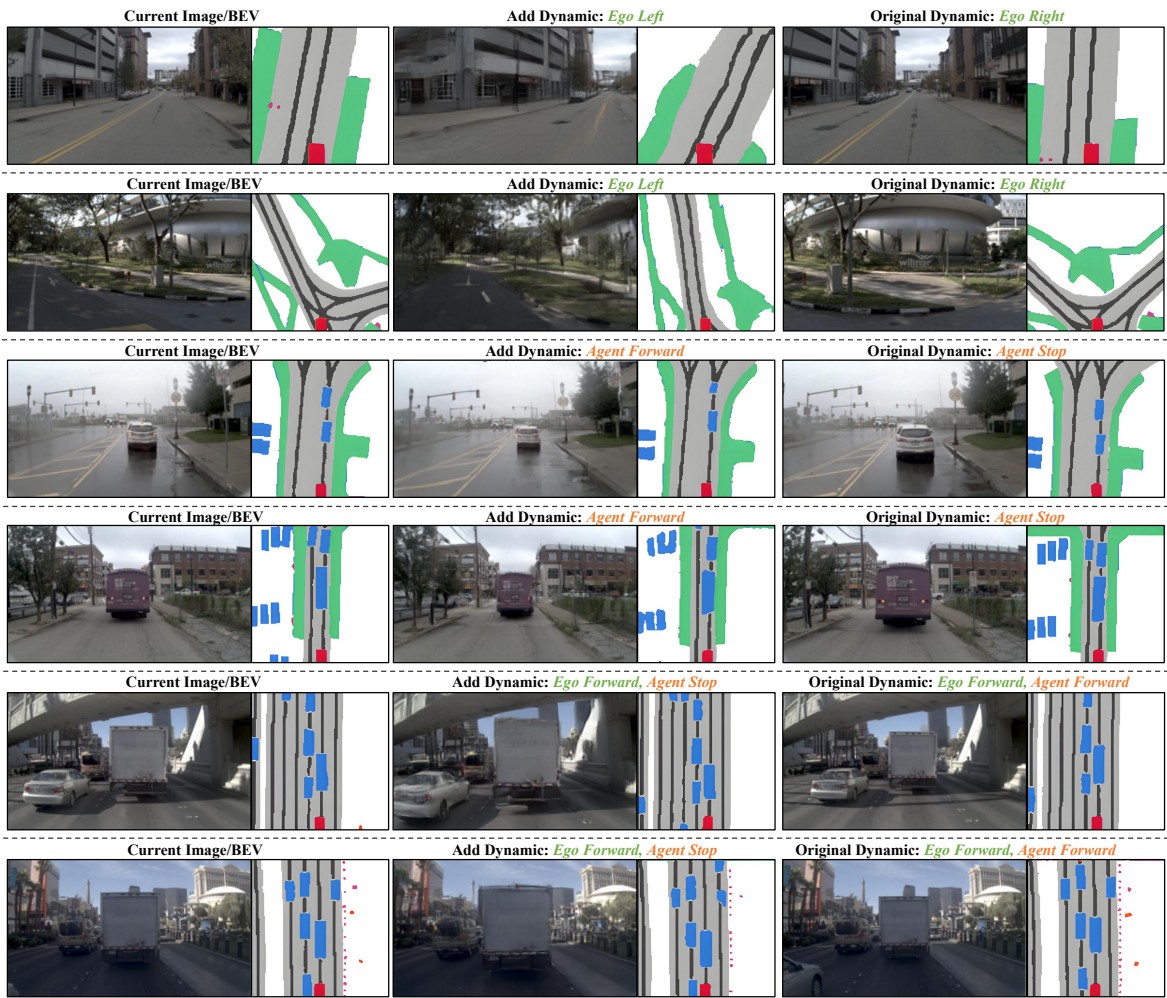

*Figure 8.* **Additional visualizations of the transferability of learned dynamics.**

while a lightweight fast planner outputs actions at every control cycle by consuming the most recent observation and reasoned dynamics tokens.

## F. Failure Cases

Our trajectory planning relies on the inferred future dynamics as intermediate reasoning signals. Consequently, when the predicted dynamics are inaccurate or ambiguous, the planned decisions may also be suboptimal. One possible failure arises from misinference about other agents' intentions. As illustrated in the first and second rows of Fig. 10, the model erroneously predicts the leading vehicle to continue moving forward and thus plans to follow with a straight trajectory. Another failure case occurs in complex turning scenarios, where the model incorrectly reasons about future road-structure dynamics. As shown in the third row of Fig. 10, the model fails to recognize a non-drivable parking area. This is because predicting road geometry that will only become visible from a novel perspective is challenging.

Finally, when the current observation is severely degraded, the inferred future dynamics can become ambiguous. As shown in the fourth row of Fig. 10, heavy rain causes camera occlusion and results in blurred observations. Under such conditions, the predicted dynamics lack sufficient certainty, and the model makes unsafe decisions.

## G. More Related Works

### G.1. Latent Action Tokenizer

Latent action tokenizers aim to overcome the scalability bottleneck of action-labeled data by extracting action-like tokens directly from raw video. LAPO (Schmidt & Jiang, 2024) shows that a latent inverse-dynamics structure can be recovered purely from videos, enabling latent-action policies. LAPA (Ye et al., 2024) explicitly learns a VQ-style discrete latent action codebook between adjacent frames and pretrains a VLA to predict these latent actions with a lightweight decoder to real robot controls.

| w/o Dyn CoT | w/ Dyn CoT | Current Observation | Dyn CoT Decoded Future |
|---|---|---|---|

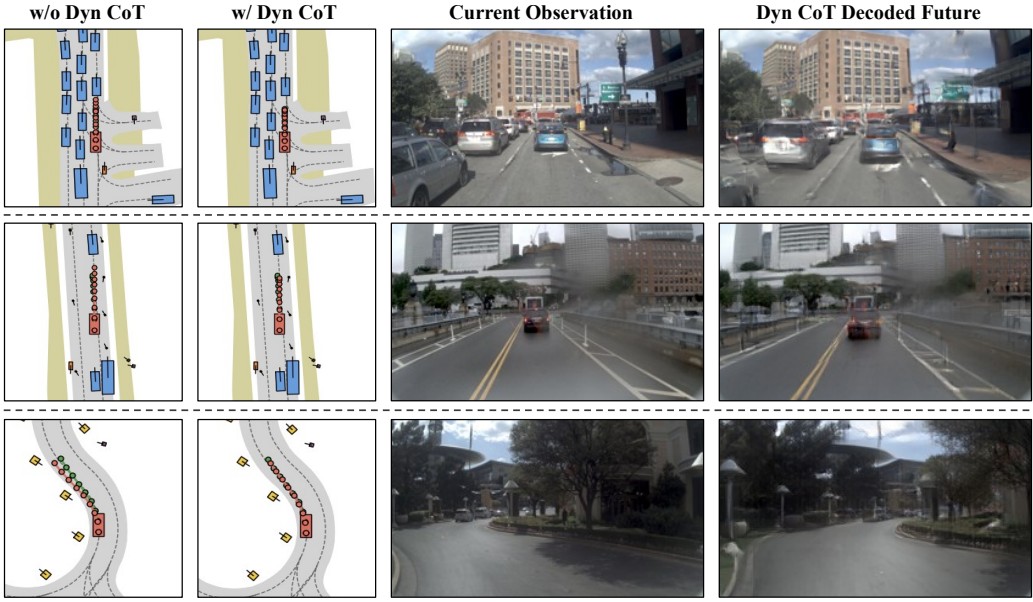

*Figure 9.* **Additional qualitative comparisons of planning behavior with and without Dynamics CoT.**

Moto (Chen et al., 2025b) treats latent action tokens as a hardware-agnostic motion language and pretrains an autoregressive model over these tokens, then co-finetunes to bridge motion-token prediction to executable robot actions. UniVLA (Bu et al., 2025) emphasizes transfer across embodiments by extracting task-centric latent actions from large-scale cross-embodiment videos and efficiently decoding them for different robots. LAOM (Nikulin et al., 2025) highlights that purely unsupervised latent action learning can be confounded by action-correlated distractors, and injecting sparse ground-truth action supervision during latent-action training substantially improves downstream control. In contrast to these works that primarily focus on learning a general latent action space for embodied settings, our goal is driving-oriented: we explicitly disentangle ego-centric and environment-centric dynamics across multi-agent, viewpoint-changing scenes and further introduce a cross-space regularization that enforces semantic alignment between dynamics learned from the image space and the BEV space.

### G.2. Foresight-driven Policies

Foresight-driven policies have been widely studied in embodied control, where actions are predicted by first anticipating future outcomes. Early work, such as HVF (Nair & Finn, 2020) decomposes tasks via visual subgoal generation, while GCPs (Pertsch et al., 2020) further adopt goal-conditioned coarse-to-fine predictors to reduce long-horizon planning complexity. More recently, foresight has been integrated into end-to-end and generalist policies. VPP (Hu et al., 2024) conditions a single policy on predictive visual representations learned from large-scale video models, and recent VLA-style methods (Tian et al., 2025; Zhao et al., 2025a; Lv et al., 2025) treat predicted future states or visual reasoning steps as intermediate signals that guide inverse dynamics for action generation. In autonomous driving, foresight-driven policies are also adopted to handle long horizons and multi-agent interactions. FSDrive (Zeng et al., 2025) introduces a spatio-temporal visual CoT by generating future visual states. PWM (Zhao et al., 2025b) proposes a policy world model that forecasts future states and performs collaborative state–action prediction to better couple forecasting and planning. However, existing approaches predominantly rely on explicit dense future-frame prediction, which introduces redundant reasoning and non-trivial inference overhead. In contrast, we focus on predicting compact future dynamics, preserving decision-critical foresight while enabling latency-efficient inference.

In addition, LAW (Li et al., 2024b) enhances end-to-end driving via a latent world model that predicts future latent features, supervised by latent representations of future observations in a self-supervised manner. SSR (Li & Cui, 2025) also exploits temporal context for self-supervision, reducing reliance on expensive perception annotations. Notably, they primarily improve driving via self-supervised learning, whereas we compress future evolution into compact dynamics tokens that serve as an explicit intermediate reasoning trace for action generation.

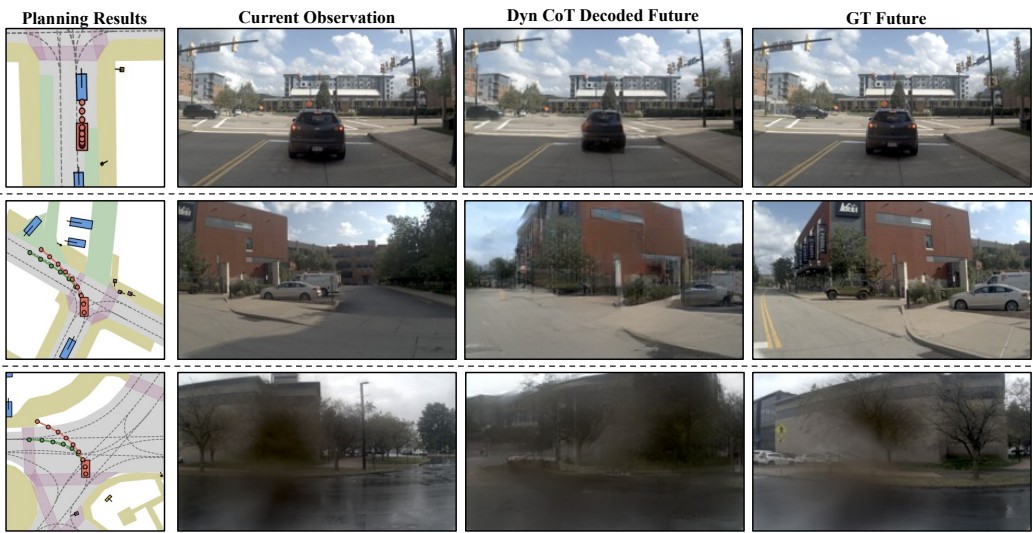

*Figure 10.* **Representative failure cases.** From left to right, we show the planning result, the current observation, the future decoded from inferred dynamics, and the ground-truth future. These cases include incorrect inference of surrounding vehicles' intentions, misidentification of drivable areas during large turning maneuvers, and ambiguous dynamics reasoning due to degraded visual observations.

