# OpenReview forum: "DynVLA: Learning World Dynamics for Action Reasoning in Autonomous Driving"
_ICML.cc/2026/Conference — ICML 2026 regular_

### Official Review · Reviewer_xBem · 2026-02-25

**Soundness:** 3
**Presentation:** 3
**Significance:** 2
**Originality:** 2
**Overall Recommendation:** 4
**Confidence:** 4

**Summary:**

This work presents DynVLA, a vision-language-action model tailored for driving scenarios. The model is equipped with a novel dynamic CoT mechanism, where ego-centric view and BEV information are jointly encoded into dynamic tokens, and are subsequently trained with a next-token prediction model. The author conducted experiments on two driving benchmarks and one in-house dataset, and did both SFT and RFT in ablation studies, demonstrating good performance transfer. Although not significant, in general, the proposed method is considered novel with minor clarifications needed, however, the vagueness of the dataset and architecture in training needs to be addressed.

**Compliance With Llm Reviewing Policy:**

Affirmed.

**Key Questions For Authors:**

1. Please clarify the in-house dataset details: how did you collect the dataset? Is it synthetic or real-world? Do you have plans to make it open source? It is advised to move some information from appendix to the main text.
2. Please clarify whether some pre-trained models are involved in the process. Are there any existing backbone or foundational models involved?
3. Please clarify if both SFT and RFT are used in producing the main experiment tables. How does it perform if only RFT is used?

**Limitations:**

Yes.

**Strengths And Weaknesses:**

Strength:
1. The author provided detailed experiments, especially in terms of models compared.
2. Merging ego-centric view with BEV is a natural and smart idea, providing more information to the model with less overhead.
3. The involvement of both SFT and RFT made the model training more flexible.

Weakness:
1. BEV mainly considers vehicles, objects and roads. Other environmental factors (weathers, sunlight, etc) that might affect planning is not involved.
2. The model architecture and dataset contribution is not clearly illustrated, limiting its novelty and less convincible.
3. The authors claim that explicit decoupling of ego-centric and environmental data contributes to understanding, yet there are limited ablation studies that supports the claim.

---

> ### Author Rebuttal · Authors · 2026-03-30
>
> Thank you for your detailed and constructive reviews. We are glad that you found our method “is considered novel” and “is a natural and smart idea”. We would like to address the weaknesses (**W**) and questions (**Q**) below.
>
> ---
> ### [W1] Other environmental factors (weather, sunlight, etc.) are not involved in BEV.
> We would like to carefully clarify that **the image branch will preserve appearance information (e.g., weather, sunlight) in the Dynamics Tokenizer.** This is because the image reconstruction loss requires the tokenizer to recover fine-grained appearance cues from the input images. Therefore, environmental factors such as weather and sunlight are naturally encoded in the learned dynamics tokens. We will include this discussion in the revised version.
>
> ---
> ### [W2.1, Q1] The dataset contribution is not clearly illustrated. How did you collect the dataset? Is it synthetic or real-world? Do you have plans to make it open source?
> We appreciate your request for clearer dataset details, which helps strengthen the transparency of our work. **We plan to release this in-house dataset**, as we believe its scale and diversity could provide valuable support in this community. This dataset is **a real-world dataset**, collected by human drivers using dedicated data-collection vehicles. It **covers 339 cities and has a total scale of 70M frames**, and is substantially larger than existing public datasets. The data **includes 8 camera views, and the image resolution is 1280×720, with a sampling frequency of 10 Hz**. In addition, we provide an anonymous link with representative examples: https://anonymous.4open.science/r/A7D2F9A304. Since the dataset is very large-scale, the release process requires additional time for data organization, anonymization, and compliance checking, we are actively preparing it for public release.
>
> ---
> ### [W2.2, Q2] The model architecture is not clearly illustrated. Whether some pre-trained models, existing backbone or foundational models are involved?
>
> We apologize for not making model architecture clear, and **will provide a clearer description of the model architecture and the backbone used in the revised paper.** We use the **pre-trained EMU3 and Qwen2.5-VL models as backbones**. EMU3 adopts a pure Transformer decoder architecture, and Qwen2.5-VL first uses a ViT to map images into the text space, and then uses multiple Transformer decoder layers to generate outputs. In addition, our **Dynamics Tokenizer is trained from scratch**. The image and BEV patchifiers are CNN-based, while both the encoder and decoder are built with multiple Transformer layers.
>
> ---
> ### [W3] Limited ablation studies to support the claim that explicit decoupling of ego-centric and environmental data contributes to understanding.
>
> Thank you for this valuable suggestion, which motivated us to further strengthen the evidence for the decoupling design. We would like to kindly clarify that **the effect of explicit decoupling has been validated in Figure 5 and Table 6 in the main paper**. To further strengthen this claim, we additionally provide two analyses and will include these in the revised version. As shown in the table below, **introducing explicit decoupling significantly improves codebook utilization.**
>
> |Tokenizer|VQ Codebook Size|Mean Activated VQ Codes ↑|Codebook Utilization Rate (%) ↑|
> |-|:-:|:-:|:-:|
> |w/o Decouple|64|9.3|14.5|
> |w/ Decouple|64|59.2|92.5|
> |w/o Decouple|128|16.1|12.6|
> |w/ Decouple|128|96.7|75.5|
>
> From the perspective of downstream planning, as shown in the table below in the SFT stage, removing decoupling brings almost no improvement, while **introducing decoupling leads to a clear performance gain**.
>
> |Method|PDMS ↑|
> |-|:-:|
> |Baseline w/o CoT|85.6|
> |Tokenizer w/o Decouple|85.8|
> |**Tokenizer w/ Decouple**|**87.2**|
>
> ---
> ### [Q3] Please clarify if both SFT and RFT are used to produce the main experiment tables. How does it perform if only RFT is used?
>
> **The results reported in the main experiment tables are obtained from the full training pipeline (first SFT, then RFT), and our method can already benefit from RFT after a relatively short SFT warm-up.** Regarding the setting of using RFT only, we would like to humbly clarify that **RFT needs to be built upon an SFT-initialized model**. This is because reinforcement learning requires the model to generate reasonable trajectories during sampling, so as to obtain non-zero rewards and form effective learning signals. However, since our initialization comes from general pre-trained VLM weights, the model does not yet possess action-generation capability without SFT.
>
> We further provide additional ablations with varying numbers of SFT steps, followed by subsequent RFT training. As shown below, **SFT does not need to be extensive, and even a short SFT warm-up enables effective gains from subsequent RFT**.
>
> |SFT Steps|SFT PDMS ↑|SFT+RFT PDMS ↑|
> |-|:-:|:-:|
> |0.5k|72.0|83.9|
> |1k|79.6|86.4|
> |2k|83.1|88.3|
> |3k|86.4|89.9|
> |4k|87.2|91.7|

---

> > ### Author Rebuttal · Reviewer_xBem · 2026-04-03
> >
> > My concerns are fully resolved.

---

> > > ### Author Response · Authors · 2026-04-04
> > >
> > > Thank you very much for your encouraging feedback and for confirming that our rebuttal has adequately addressed your concerns. We greatly appreciate your recognition of our work, as well as the valuable comments that helped us further strengthen the paper.

---

### Official Review · Reviewer_Z59e · 2026-02-27

**Soundness:** 2
**Presentation:** 3
**Significance:** 3
**Originality:** 3
**Overall Recommendation:** 4
**Confidence:** 3

**Summary:**

The paper tackles the inefficiency and representational limits of current Vision-Language-Action (VLA) models in autonomous driving. Specifically, it addresses the flaws in existing Chain-of-Thought (CoT) paradigms: Textual CoT lacks fine-grained spatiotemporal understanding for physical environments, while Visual CoT introduces massive computational overhead and redundancy by forcing the model to generate decision-irrelevant background pixels. The authors propose DynVLA, a model that introduces Dynamics CoT. Instead of generating text or full images, the model uses a Dynamics Tokenizer to compress future world evolution into a tiny set of discrete, vector-quantized tokens.

**Compliance With Llm Reviewing Policy:**

Affirmed.

**Final Justification:**

Initially, my primary concern regarding the framework's soundness and practical significance was the prohibitive inference latency, which undermined real-world safety. However, the authors' rebuttal addressed this by introducing a query-based Dynamics CoT that successfully reduces latency to real-time levels (~10 FPS) without sacrificing the reasoning capacity of the larger 7B backbone. I am raising my score accordingly.

**Key Questions For Authors:**

With an inference latency of 0.37s on an H800 GPU, at what point does the benefit of CoT get completely neutralized by the physical distance the car travels while the model is still thinking?

**Limitations:**

The authors do provide a limitaion section in the appendix, but it does not cover the points mentioned in the above sections of the review.

**Strengths And Weaknesses:**

Conceptually, the work is significant because it challenges the assumption that VLA models must reason using human-interpretable modalities like text or video. However, while the authors claim that Dynamics CoT drops latency compared to Visual CoT, a 370ms inference delay on a top-tier GPU means less than 3 frames per second. In the context of real-world, safety-critical autonomous driving, a 370ms delay is still dangerous, undermining the practical value of the work claimed by the authors. Furthermore, the performance of DynVLA on NAVSIM Benchmark is not impressive, and the metrics of the other benchmarks are not as comprehensive as NAVSIM Benchmark. Therefore, the effectiveness of DynVLA is also questionable.

---

> ### Author Rebuttal · Authors · 2026-03-30
>
> Thank you for your detailed and constructive reviews. We are glad that you found our method “is conceptually significant because it challenges the assumption that VLA models must reason using human-interpretable modalities”. We would like to address the weaknesses (**W**) and questions (**Q**) below.
>
> ---
> ### [W1] A 370 ms delay is still dangerous, undermining the practical value.
>
> Thank you for highlighting this important practical concern, which helps us improve the real deployment implications of our method. We agree that 0.37s is still too high for direct real-world use, and we show that **this latency can be mitigated through smaller models and deployment optimizations**.
>
> We would first like to carefully clarify that in practical deployment, techniques such as vLLM, operator-level optimizations, and quantization can further reduce latency. As shown in the table below, we provide additional results **using vLLM acceleration and smaller VLM backbones**. To better reflect realistic deployment conditions, **we test on the NVIDIA RTX 4090 GPU** instead of top-tier GPUs. The results show that our method maintains competitive performance while significantly reducing inference latency. For example, Qwen3VL-2B achieves **0.07s latency (\~13 FPS)**, which is compatible with real-time planning requirements.
>
> Furthermore, we compare with recent VLA methods. Even **without vLLM**, our method (Qwen2.5VL-7B) already **achieves lower latency with better performance**. This advantage becomes more pronounced with vLLM acceleration. We will include these additional experiments and discussion in the revised version.
> |Model|Ref|Latency (s) ↓|GPU|PDMS ↑|
> |-|-|:-:|:-:|:-:|
> |AutoVLA|NeurIPS’25|1.31|NVIDIA L40|89.1|
> |PWM|NeurIPS’25|0.85|NVIDIA A800|88.1|
> |AutoDrive-R²|ICLR’26|3.62|NVIDIA H20|90.3|
> |Curious-VLA|CVPR’26|1.57|NVIDIA H100|90.3|
> |AdaThinkDrive|ICRA’26|0.74|NVIDIA H20|90.3|
> |**Ours (Qwen2.5VL-7B) w/o vLLM**|/|**0.48**|**RTX 4090**|**91.0**|
> |**Ours (Qwen2.5VL-7B) w/ vLLM**|/|**0.22**|**RTX 4090**|**91.0**|
> |**Ours (Qwen2.5VL-3B) w/ vLLM**|/|**0.13**|**RTX 4090**|**90.3**|
> |**Ours (Qwen3VL-4B) w/ vLLM**|/|**0.15**|**RTX 4090**|**90.5**|
> |**Ours (Qwen3VL-2B) w/ vLLM**|/|**0.07**|**RTX 4090**|**90.0**|
>
> ---
> ### [W2] The performance of DynVLA on the NAVSIM benchmark is not impressive.
>
> We would like to humbly clarify that **the improvement of DynVLA on the NAVSIM benchmark is non-trivial relative to recent reported advances**. As shown in the table below, we summarize the final PDMS of several recent works together with **their reported PDMS gains over the previous SOTA in their respective papers**. Under this comparison, **DynVLA achieves a larger improvement than these recent methods**, suggesting that our gain on NAVSIM is not marginal.
>
> We would also like to emphasize that **the primary contribution of our work is to model a compact future dynamics CoT, as an alternative to conventional textual or visual CoT**. As shown in Table 4 of the main paper, our method achieves both improved performance and reduced CoT latency compared to other CoT paradigms, which directly supports the effectiveness of our proposed reasoning design.
>
> |Model|Ref|PDMS ↑|Reported Gain over Previous SOTA ↑|
> |-|-|:-:|:-:|
> |BridgeDrive|ICLR’26|88.0|0.4|
> |DriveVLA-W0|ICLR’26|90.2|0.6|
> |VADv2|ICLR’26|89.3|0.7|
> |Curious-VLA|CVPR’26|90.3|0.7|
> |MeanFuser |CVPR’26|89.0|0.7|
> |**Ours (Qwen2.5VL)**|/|**91.0**|**1.0**|
> |**Ours (EMU3)**|/|**91.7**|**1.7**|
>
> ---
> ### [Q1] With a latency of 0.37s on an H800 GPU, at what point does the benefit of CoT get completely neutralized by the physical distance the car travels while the model is still thinking?
>
> Thank you for this insightful question, which highlights the important trade-off between reasoning latency and physical execution in real-world driving. **Under more realistic deployment settings and smaller backbones, the latency overhead of Dynamics CoT can be mitigated**. Moreover, **CoT can be selectively activated only in scenarios where reasoning is needed**; in high-speed scenarios such as highway driving, CoT can be optionally disabled.
>
> We would first like to kindly clarify that this latency is measured without modern inference optimizations. In practical deployment, **techniques such as vLLM, operator-level optimizations, and quantization can substantially reduce inference time**. In addition, **using smaller VLM backbones provides a favorable speed–performance trade-off.** As shown in the table in W1, **on the RTX 4090 GPU**, Qwen3VL-2B with vLLM achieves **\~0.07s latency (\~13 FPS)** while maintaining strong planning performance.
>
> We also want to note that the benefit of Dynamics CoT is pronounced in complex, interactive, and long-horizon situations. In simpler or high-speed scenarios where the distance traveled during CoT reasoning may offset its benefits, we can disable CoT and use lighter planning modes. We will clarify this trade-off in the revised version.

---

> > ### Author Rebuttal · Reviewer_Z59e · 2026-04-03
> >
> > Thank you for providing the additional latency experiments and the benchmark context. While I acknowledge that using a smaller 2B parameter model and vLLM reduces latency to 0.07s, this inevitably trades off the very reasoning capacity your framework seeks to demonstrate. Furthermore, your suggestion to selectively disable the Dynamics CoT in high-speed scenarios is a highly concerning workaround. High-speed environments are precisely where foresighted, safety-critical reasoning is most paramount. If the proposed framework is too slow to be safely utilized in these conditions and must simply be turned off, its practical value for real-world autonomous driving remains fundamentally constrained. Because this core tension between reasoning overhead and real-time execution in safety-critical scenarios is not fully resolved, I am maintaining my current score.

---

> > > ### Author Response · Authors · 2026-04-04
> > >
> > > We sincerely appreciate your concern and agree that reasoning overhead versus real-time execution is a critical issue in safety-critical scenarios. You pointed out a core and highly valuable issue regarding real-world deployment, which further motivated us to explore methods to further reduce the latency of Dynamics CoT. Our new results show that, by adopting a **query-based Dynamics CoT, a 7B model can retain the benefits of Dynamics CoT while achieving \~10 FPS**, making it more feasible to **apply CoT under real-time constraints, even in high-speed scenarios**. We hope the additional analysis below helps address your concern, and we sincerely welcome further discussion on this important point.
> > >
> > > We agree that using smaller models may weaken the reasoning capacity that our framework aims to demonstrate. Therefore, instead of relying on smaller backbones, we focus on **reducing the latency of the 7B model itself**. To this end, we explore a **query-based Dynamics CoT** design. Instead of autoregressively generating CoT tokens, we use a small set of learnable queries placed before action tokens to directly perform dynamics reasoning. As a result, the CoT process no longer incurs step-by-step decoding overhead, and only introduces a small number of prefill tokens. This design **significantly reduces the additional decoding latency introduced by CoT**.
> > >
> > > As shown in the table below, we observe that **query-based Dynamics CoT maintains comparable performance to standard Dynamics CoT while introducing almost no additional latency** (0.10s vs. 0.09s). This allows the **7B model** to run at **~10 FPS without disabling CoT**. We believe this query-based Dynamics CoT can substantially **alleviate the tension between reasoning overhead and real-time execution**.
> > >
> > > | Model                                   | Latency (s) ↓ | FPS ↑ | PDMS ↑ |
> > > | --------------------------------------- | :-----------: | :---: | :----: |
> > > | Qwen25VL-7B w/o Dynamics CoT            |     0.09      |  11   |  88.4  |
> > > | Qwen25VL-7B w/ Dynamics CoT             |     0.22      |   4   |  91.0  |
> > > | Qwen25VL-7B w/ query-based Dynamics CoT |     0.10      |  10   |  90.7  |
> > >
> > > Finally, we would like to more carefully clarify the scope of our claims. Our goal is to show that, **compared to existing textual or visual CoT approaches, Dynamics CoT significantly reduces reasoning redundancy and advances the efficiency–performance trade-off** (as shown in the table in W1). The proposed query-based variant further indicates that this compact dynamics reasoning form can be pushed closer to real-time deployment. We will revise the paper to include this query-based design, together with its experimental results and deployment-oriented analysis.

---

### Official Review · Reviewer_jEdW · 2026-03-11

**Soundness:** 2
**Presentation:** 3
**Significance:** 2
**Originality:** 3
**Overall Recommendation:** 4
**Confidence:** 4

**Summary:**

This paper proposes a CoT paradigm for VLA-based autonomous driving, by introducing dynamic tokens to enhance model's spatiotemporal understanding capabilities while reducing redundancy and latency compared to text-based or image-based CoT methods. The paradigm includes a Dynamic Tokenizer that learns ego-centric and environment-centric tokens in a disentangled way through ego action and cross-view consistency regularization.

After training the tokenizer, the paper applies standard SFT and RFT to improve Dynamics CoT performance. Results on NAVSIM and Bench2Drive validate the model performance in both real-world and closed-loop settings. Ablation studies are conducted to validate the effectiveness of the design choices.

**Compliance With Llm Reviewing Policy:**

Affirmed.

**Final Justification:**

The rebuttal has addressed most of my concerns, with one caveat (see my notes below). Thus I'm raising my score.

**Key Questions For Authors:**

* See my questions in the Weaknesses section.
* What's K in Figure 2(b)? Why is that different from N (action horizon)?
* In Figure. 4, how are the latent dynamic codes mapped to explicit dynamics ("Ego Right", "Agent Forward")?
* Have the authors study the effect of different VLA model parameter sizes? Is the proposed method limited to a minimum VLA size?

I'm willing to adjust my scores based on the responses to these questions, especially the ones in the Weaknesses section.

**Limitations:**

Yes

**Strengths And Weaknesses:**

Strengths:
* The idea of introducing latent dynamic tokens is novel. Compared to explicit text and visual tokens, latent tokens allow more efficient reasoning for VLAs. As latency remains a main bottleneck to deploy VLA in real-time, this paper proposes a promising direction to mitigate it.
* The design choices are intuitive and validated in experiments. Although the paper could benefit more from more justifications and clarifications on some of them (see Weaknesses below)
* The paper is written clearly, with figures that are easy to follow and detailed experimental results.
* The results demonstrate the strong performance of the proposed method across two benchmarks.

Weaknesses:
* The paper spends a lot of space showing the results, yet lacks a formal problem statement and implementation details. While the authors include some of these in Appendix, it is a common practice to present a brief statement and summary of implementation details in the main paper.
* The inputs to the model remains unclear to me. During SFT training, the model takes the current image observation plus the previous observation. Is there a reason not to take more history frames? Does the input include a single-view or multiple-view? Can the authors provide justifications on these design choices?
* The cross-view consistency regularization deserves more clarifications. How could the model ensures the ego-view is consistent with the BEV view? For instance, if an agent turns left in the ego-view, does the model enforce that agent exhibits the same behavior in the BEV view?
* The paper reports latency in Table 4, yet 0.37s still remains a big concern for real use cases. Have the authors tried smaller VLA variants?
* Can the author provides more details of Driving Simulator in RFT training?
* The in-house dataset results do not provide much value, without disclosing any details on the dataset (i.e. their sizes, data distributions, examples, etc.) The authors should consider adding more details or moving these to Appendix.

---

> ### Author Rebuttal · Authors · 2026-03-30
>
> Thank you for your detailed and constructive reviews. We are glad that you found our method “the idea is novel” and “proposes a promising direction”. We would like to address the weaknesses (**W**) and questions (**Q**) below.
>
> ---
> ### [W1] Lacks a formal problem statement and implementation details.
> Thank you for pointing out this important omission. Formally, given current and historical observations $o_{t-H:t}$, the goal is to predict a future ego trajectory $y_{t+1:t+N}$, where $N$ is the planning horizon and $y_{t+i}$ denotes the waypoint at step $t+i$. We also summarize some key implementation details in table below. **We will include a formal problem statement and key implementation details in the revised main paper**.
> |Training Stage|Steps|GPU|Learning rate|Batch size|
> |-|:-:|:-:|:-:|:-:|
> |Tokenizer|200k|8×L20|1e-4|32|
> |SFT|4k|8×L20|1e-4|6|
> |RFT|6k|6×H800|2e-6|6|
>
> ---
> ### [W2] The inputs to the model remains unclear.
> **We use one historical observation from 1s earlier together with multi-view image inputs** (front, left, and right). As shown in the table below in the SFT stage, adding an additional historical observation from 2s earlier may introduce outdated temporal cues, which hurts performance, while multi-view inputs provide richer information and improve performance.
>
> |History Frame|Views|PDMS ↑|
> |:-:|:-:|:-:|
> |**1s**|**Multi-view**|**86.6**|
> |1s, 2s|Multi-view|85.8|
> |1s|Single-view|86.2|
>
> ---
> ### [W3] The cross-view consistency regularization deserves more clarifications.
> Our method **does not impose a hard constraint between ego-view and BEV**. Instead, **it uses a shared-representation regularization during training**: the same dynamics tokens are fed into two decoders to predict future image and future BEV, each supervised by its own target. Since both targets describe the same future scene in different views, this encourages the tokens to capture semantically consistent dynamics across views.  We will clarify this regularization in the revised version.
>
> ---
> ### [W4] Latency still remains a concern. Have smaller VLA variants been tried?
> Thank you for this helpful suggestion, which makes our latency-performance analysis more complete. As shown in the table below, **our approach maintains strong planning performance while reducing latency with smaller models**. Tested on the deployment-friendly RTX 4090 GPU, Qwen3VL-2B reaches only **0.07s latency**. We will include these experiments in the revised version.
> |Model|Latency (s) ↓|PDMS ↑|
> |-|:-:|:-:|
> |Qwen3VL-2B|0.07|90.0|
> |Qwen25VL-3B|0.13|90.3|
> |Qwen3VL-4B|0.15|90.5|
>
> ---
> ### [W5] Provide more details of Driving Simulator in RFT.
> We use a **non-reactive driving simulator** as in NAVSIM. The agents follow their future trajectories recorded in the logs, while the ego vehicle is advanced according to the predicted trajectory. We will provide a clearer description in the revised version.
>
> ---
> ### [W6] Adding more details about the in-house dataset.
> Thank you for pointing this out, which helps make our in-house dataset results more informative and convincing. Our in-house dataset **covers 339 cities, with a total scale of 70M frames**, and is substantially larger than existing public datasets. The data **contains 8 camera views with 1280×720 image resolution and 10 Hz sampling frequency**. We also provide several representative scenario examples in https://anonymous.4open.science/r/A7D2F9A304. **We will provide more details about the in-house dataset in the revised version.**
>
> ---
> ### [Q1] What's K in Figure 2(b) and its difference between N (action horizon)?
> **K in Figure 2(b) denotes the predicted future dynamics steps before action generation**, each step corresponds to 1s interval and we use K = 2. In contrast, N represents the number of action steps. We will clarify the definition of K in the caption of Figure 2.
>
> ---
> ### [Q2] In Figure 4, how are the dynamics codes mapped to explicit dynamics ?
> We first **manually select source scenarios with clear dynamics patterns**, such as the ego moving right or a agent moving forward, and then **encode these scenarios to obtain the corresponding dynamics tokens**. These extracted tokens are then injected into a new scenario for future-state decoding.
>
> ---
> ### [Q3] The effect of proposed method in different model sizes?
> Thank you for this constructive question, which motivates a more thorough scaling analysis. **Our method can bring improvements across different model scales.** As shown in the table below, we compare different backbone sizes with and without Dynamics CoT, and will include these results in the revised version.
> |Model|PDMS (SFT) ↑|PDMS (SFT + RFT) ↑|
> |-|:-:|:-:|
> |Qwen25VL-3B w/o Dyn CoT|84.8|87.8|
> |Qwen25VL-3B w/ Dyn CoT|85.8|90.3|
> |Qwen3VL-2B w/o Dyn CoT|84.0|87.4|
> |Qwen3VL-2B w/ Dyn CoT|85.6|90.0|
> |Qwen3VL-4B w/o Dyn CoT|85.1|88.0|
> |Qwen3VL-4B w/ Dyn CoT|86.1|90.5|

---

> > ### Author Rebuttal · Reviewer_jEdW · 2026-04-03
> >
> > The rebuttal has resolved most of my concerns.
> >
> > I'm still not fully convinced by the claim that "adding an additional historical observation from 2s earlier may introduce outdated temporal cues". In some challenging driving cases (i.e. stop signs, unprotected left turns), longer historical cue is crucial to understanding the temporal context. I'd encourage the authors to provide more in-depth analysis of this regression.

---

> > > ### Author Response · Authors · 2026-04-03
> > >
> > > Thank you for your acknowledgement of our rebuttal and for increasing your score. We also sincerely appreciate this insightful follow-up question, which has further inspired our thinking on future work. We agree that **longer temporal context can be important in temporally demanding scenarios**, and this is also supported by our additional analysis below: **in temporally demanding cases, we observe that incorporating longer history is indeed beneficial**. We apologize for the lack of clarity in our previous wording. Our conclusion is not that longer history is generally unhelpful, but rather that under our current model design, naively adding an additional raw observation from 2s earlier does not bring consistent performance gains.
> > >
> > > This observation is also **consistent with findings in prior end-to-end driving papers and recent VLA-based approaches**: TransFuser [1] notes that *"we use a single time-step input since prior works on imitation learning for autonomous driving have shown that using observation histories may not lead to performance gain".* Similarly, DriveVLA-W0 [2] observes that "*using temporally distant historical frames may introduce excessive scene variation, making planning more challenging*". Moreover, recent end-to-end driving methods, such as DiffusionDrive [3], ReCogDrive [4], and WoTE [5], also do not rely on extended observation histories.
> > >
> > > We believe a possible explanation is that **different scenarios require different amounts of temporal context**. In highly complex scenarios, longer history can indeed be beneficial, while for more common cases, a 1s historical observation is often sufficient. We further conduct a targeted case-based analysis by manually selecting and grouping representative evaluation cases into two categories: **temporally demanding cases** (e.g., unprotected left turns, stop signs, and traffic-light interactions) and **simple cases** (e.g., straight driving, cruising, and simple following). As shown in the table below, **adding a 2s historical observation improves performance in temporally demanding cases.** In contrast, for simple cases, the additional 2s frame causes slight performance degradation. This suggests that the value of longer history is scenario-dependent: it is useful when the decision depends on longer-horizon temporal evolution, but may be redundant in simpler cases. **Since these temporally demanding cases constitute a smaller portion compared to simple scenarios, their local gains do not translate into overall performance improvements.**
> > >
> > > |History Frame|Temporally Demanding Case PDMS ↑|Simple Case PDMS ↑|Overall PDMS ↑|
> > > |-|:-:|:-:|:-:|
> > > |1s|84.9|**88.8**|**86.6**|
> > > |1s, 2s|**85.4**|88.2|85.8|
> > >
> > > We acknowledge that **a sufficiently strong VLA model should be able to adaptively leverage longer history in complex scenarios while ignoring redundant information in simpler ones**. However, current approaches have not yet fully achieved this capability. We believe a key reason is that **existing VLA backbones generally rely on relatively simple temporal fusion mechanisms**, which may be **insufficient to adaptively select and utilize historical information according to the demands of different scenarios**. As also noted in ReasonNet [6], while temporal context is important, naive historical fusion without dedicated temporal reasoning or alignment mechanisms is often suboptimal.
> > >
> > > From the above experiments and analysis, we believe **a promising future direction is to explore adaptive history selection in VLA models, enabling the model to utilize longer temporal context when necessary**. We will include a more detailed discussion on the role of historical frames in the revised version, and further elaborate on the limitations of our current temporal modeling design as well as possible future directions.
> > >
> > > [1] Chitta, Kashyap, et al. "Transfuser: Imitation with transformer-based sensor fusion for autonomous driving." IEEE transactions on pattern analysis and machine intelligence 45.11 (2022): 12878-12895.
> > >
> > > [2] Li, Yingyan, et al. "DriveVLA-W0: World models amplify data scaling law in autonomous driving." arXiv preprint arXiv:2510.12796 (2025).
> > >
> > > [3] Liao, Bencheng, et al. "Diffusiondrive: Truncated diffusion model for end-to-end autonomous driving." *Proceedings of the Computer Vision and Pattern Recognition Conference*. 2025.
> > >
> > > [4] Li, Yongkang, et al. "Recogdrive: A reinforced cognitive framework for end-to-end autonomous driving." *arXiv preprint arXiv:2506.08052* (2025).
> > >
> > > [5] Li, Yingyan, et al. "End-to-end driving with online trajectory evaluation via bev world model." *Proceedings of the IEEE/CVF International Conference on Computer Vision*. 2025.
> > >
> > > [6] Shao, Hao, et al. "Reasonnet: End-to-end driving with temporal and global reasoning." Proceedings of the IEEE/CVF conference on computer vision and pattern recognition. 2023.

---

### Official Review · Reviewer_WqHv · 2026-03-11

**Soundness:** 3
**Presentation:** 3
**Significance:** 3
**Originality:** 2
**Overall Recommendation:** 4
**Confidence:** 4

**Summary:**

The authors propose DynVLA an end to end model for autonomous driving. The core idea is to learn world dynamics using a dynamic visual tokenizer. This approach aims to reduce computational cost while maintaining high driving performance. The method is evaluated on standard benchmarks and shows promising results.

**Compliance With Llm Reviewing Policy:**

Affirmed.

**Final Justification:**

The rebuttal reinforced my prior assessment and therefore I will keep my original score.

**Key Questions For Authors:**

N/A

**Limitations:**

yes. But the authors should also comment on the potential overhead or limitations of their specific decoupled tokenizer design in real world deployments.

**Strengths And Weaknesses:**

### Strengths
* **Efficient Foresight**: The transition from dense video prediction to compact dynamics tokens is a significant step toward making world-model-based VLA practical for real-time robotics.
* **Structural Decoupling**: Separating ego-motion from environmental dynamics (pedestrians, other vehicles) is physically grounded and helps the model better capture the nuances of traffic interactions.
* **Empirical Performance**: The model achieves state-of-the-art results on the Bench2Drive leaderboard, demonstrating its robustness in diverse and challenging driving scenarios.
* **Clarity**: The motivation is well-articulated, and the shift from "what the scene looks like" to "how the scene moves" as a reasoning trace is a compelling narrative.

### Weaknesses
* **Missing Ablations on Tokenizer**: The paper lacks a rigorous comparison between the proposed Dynamics Tokenizer and standard alternatives like a vanilla VQ-VAE or latent world models (e.g., as seen in LAW). It is not fully clear how much of the gain comes from the specific tokenizer design versus the overall CoT framework.
* **Dataset Specification**: There is some ambiguity regarding the exact data splits and environmental configurations used for training on Bench2Drive. This makes it difficult for other researchers to precisely replicate the reported success rates.

Summary: DynVLA presents a contribution to the field of efficient world modeling for autonomous driving. The Dynamics CoT approach is logically sound and yields impressive results on competitive benchmarks. While more thorough ablations on the tokenizer and better dataset documentation are needed, the overall quality and potential impact of the work meet the requirements for acceptance.

---

> ### Author Rebuttal · Authors · 2026-03-30
>
> Thank you for your detailed and constructive reviews. We are glad that you found our method “is a compelling narrative” and  “a significant step toward making world-model-based VLA practical”. We would like to address the weaknesses (**W**) and limitations (**L**) below.
>
> ---
> ### [W1]  Lacks a comparison between the proposed Dynamics Tokenizer and standard alternatives. Unclear how much of the gain comes from the specific tokenizer design versus the overall CoT framework.
>
> Thank you for this helpful suggestion that makes our ablation study more complete. **The primary performance gain comes from our specific tokenizer design.** As shown in Table 4 of the paper, the overall CoT framework is indeed effective, as using alternative CoT content also brings performance improvements. But **the gains are significantly smaller than those achieved by our tokenizer design**.
>
> To further clarify this point, we compare several alternatives within the broader CoT paradigm in the SFT stage under the same backbone, data, and training recipe. For **Vanilla VQ-VAE CoT**, we compress future images with a vanilla VQ-VAE and directly use the resulting VQ tokens as the CoT content. For **LAW-style latent future CoT**, we follow DreamVLA [1] and perform query-based latent future prediction before planning. Beyond CoT-based alternatives, we further examine **LAW-style representation learning** in our framework by treating backbone image features as visual latents and training a latent world model to predict future visual latents. As shown in the table below, **all three alternatives bring smaller gains than our method**. We will include these additional ablation results in the revised version.
> |Category|Method| PDMS ↑ |
> |-|-| :-: |
> |–|Baseline w/o CoT | 85.6 |
> | Representation learning | LAW-style representation learning | 85.8 |
> | CoT alternative | Vanilla VQ-VAE CoT | 86.0 |
> | CoT alternative | LAW-style latent feature CoT | 86.1 |
> | **CoT alternative** | **Dynamics CoT (Ours)** | **87.2** |
>
> ---
> ### [W2] Ambiguity regarding the data splits and environmental configurations on Bench2Drive.
>
> We apologize that the data splits and environmental settings on Bench2Drive were not described clearly enough. Specifically, we use Bench2Drive v-0.0.3, with CARLA 0.9.15 as the underlying simulator. The evaluation set contains 220 short routes of approximately 150m each, covering Town01\~Town15 under diverse weather conditions. For the training data, we follow the same setup as SimLingo [2], using the PDM-Lite expert in CARLA to collect data. The training routes mainly include the Town01\~Town10 routes from TransFuser, and the Town12 and Town13 routes from the official CARLA Leaderboard 2.0. This results in approximately 3.1M training samples. For the implementation environment, we use PyTorch 2.2.0 and Transformers 4.46.3. **We will provide a more complete description of the Bench2Drive experimental setup in the revised version, and also plan to release our DynVLA code** to provide a more complete list of environment configurations.
>
> ---
> ### [L1] Should discuss the overhead of the decoupled tokenizer in real-world deployments.
>
> Thank you for pointing out this practically relevant concern, which helps us clarify the deployment cost more explicitly. We analyze the CoT latency during deployment and show that **this additional overhead can be mitigated through smaller models and deployment optimizations**. We will also provide a more explicit discussion of the potential overhead in the revised version.
>
> We would first like to carefully clarify that **the tokenizer itself is not executed as an additional module during deployment**. It is mainly used during training to construct structured dynamics supervision. Therefore, at inference time, **the additional overhead primarily comes from generating a small number of CoT tokens, rather than running the tokenizer online**. As shown in Table 4 of the paper, the added latency is approximately 0.17s. Compared to textual CoT or visual CoT, our method keeps this overhead relatively low.
>
> To further reduce deployment cost, **we explore using smaller VLM backbones and vLLM acceleration**. We also benchmark our method **on the deployment-friendly RTX 4090 GPU**. As shown in the table below, our approach maintains strong planning performance while significantly reducing inference latency. For example, Qwen3VL-2B achieves 13 FPS, which meets real-time planning requirements.
>
> |Model|Latency (s) ↓|PDMS ↑|
> |-|:-:|:-:|
> |Qwen3VL-2B|0.07|90.0|
> |Qwen25VL-3B|0.13|90.3|
> |Qwen3VL-4B|0.15|90.5|
> |Qwen25VL-7B|0.22|91.0|
>
> [1] Zhang, Wenyao, et al. "DreamVLA: A Vision-Language-Action Model Dreamed with Comprehensive World Knowledge." The Thirty-ninth Annual Conference on Neural Information Processing Systems.
>
> [2] Renz, Katrin, et al. "Simlingo: Vision-only closed-loop autonomous driving with language-action alignment." Proceedings of the Computer Vision and Pattern Recognition Conference. 2025.

---

> > ### Author Rebuttal · Reviewer_WqHv · 2026-04-03
> >
> > My concerns have been adequately addressed.

---

> > > ### Author Response · Authors · 2026-04-04
> > >
> > > We sincerely thank you for your positive feedback and for confirming that your concerns have been fully addressed. We also truly appreciate your recognition of our work and the opportunity to further improve it through your valuable comments.

---

### Decision · Program_Chairs · 2026-04-30

**Decision:**

Accept (regular)

**Comment:**

This paper received a mixed set of initial reviews. Reviewers generally agreed that the paper proposes a novel and practically meaningful idea by using compact dynamics tokens as CoT for driving VLAs; and found the empirical results promising. The main concerns in the first round were about clarity and completeness, including the lack of a clear problem formulation, limited implementation details, insufficient evidence for some design choices, incomplete benchmark and dataset descriptions, and questions about whether the reported latency was practical for real-time deployment.

The rebuttal helped address these concerns well. The authors clarified the model inputs, training setup, regularization design, and benchmark details, and also provided additional results on tokenizer alternatives, model scaling, latency trade-offs, and dataset statistics. Importantly, after the rebuttal and follow-up discussion, two reviewers explicitly increased their scores, while the others either indicated that their concerns had been addressed or remained supportive overall. The discussion therefore moved from a borderline assessment to a positive consensus.

Overall, I recommend accept. The paper makes a meaningful contribution toward more efficient reasoning in autonomous driving VLAs, and the additional evidence provided during the discussion strengthens the case for acceptance. For the camera-ready version, the authors should incorporate the promised clarifications and new results, especially on the formal problem setup, key implementation details, reproducibility of the benchmark configuration, stronger ablations for the tokenizer and decoupling design, and a more careful discussion of latency and deployment limitations without overclaiming real-world readiness。